# Detecting nuance in conspiracy discourse: Advancing methods in infodemiology and communication science with machine learning and qualitative content coding

**Michael Robert Haupt**[1,2]*, **Michelle Chiu**[3], **Joseline Chang**[4], **Zoe Li**[2,5], **Raphael Cuomo**[6], **Tim K. Mackey**[5,7]

1 Department of Cognitive Science, University of California San Diego, La Jolla, California, United States of America, 2 Global Health Policy & Data Institute, San Diego, California, United States of America, 3 Department of Psychology, Temple University, Philadelphia, Pennsylvania, United States of America, 4 Rady School of Management, University of California San Diego, La Jolla, California, United States of America, 5 S-3 Research, San Diego, California, United States of America, 6 Department of Anesthesiology, University of California, San Diego School of Medicine, San Diego, California, United States of America, 7 Global Health Program, Department of Anthropology, University of California, San Diego, California, United States of America

* mhaupt@ucsd.edu

**Data Availability Statement:** The full dataset collected from Twitter and the top retweeted tweets selected for content coding can be found at the

## Abstract

The spread of misinformation and conspiracies has been an ongoing issue since the early stages of the internet era, resulting in the emergence of the field of infodemiology (i.e., information epidemiology), which investigates the transmission of health-related information. Due to the high volume of online misinformation in recent years, there is a need to continue advancing methodologies in order to effectively identify narratives and themes. While machine learning models can be used to detect misinformation and conspiracies, these models are limited in their generalizability to other datasets and misinformation phenomenon, and are often unable to detect implicit meanings in text that require contextual knowledge. To rapidly detect evolving conspiracist narratives within high volume online discourse while identifying nuanced themes requiring the comprehension of subtext, this study describes a hybrid methodology that combines natural language processing (i.e., topic modeling and sentiment analysis) with qualitative content coding approaches to characterize conspiracy discourse related to 5G wireless technology and COVID-19 on Twitter (currently known as 'X'). Discourse that focused on correcting 5G conspiracies was also analyzed for comparison. Sentiment analysis shows that conspiracy-related discourse was more likely to use language that was analytic, combative, past-oriented, referenced social status, and expressed negative emotions. Corrections discourse was more likely to use words reflecting cognitive processes, prosocial relations, health-related consequences, and future-oriented language. Inductive coding characterized conspiracist narratives related to global elites, anti-vax sentiment, medical authorities, religious figures, and false correlations between technology advancements and disease outbreaks. Further, the corrections discourse did not address many of the narratives prevalent in conspiracy conversations. This paper aims to further bridge the gap between computational and qualitative methodologies

following link on the Open Science Framework (OSF): https://bit.ly/5G_Conspiracies (DOI 10.17605/OSF.IO/YRNMX). Usernames, links to the tweet, and all personally identifiable information were removed to preserve anonymity.

**Funding:** The author(s) received no specific funding for this work.

**Competing interests:** Authors TKM and ZL are employees of the startup company S-3 Research LLC. S-3 Research is a startup funded and currently supported by the National Institutes of Health – National Institute of Drug Abuse through a Small Business Innovation and Research contract for opioid-related social media research and technology commercialization. TKM is also the CEO and a member of S-3 Research LLC with ownership. Author reports no other conflict of interest associated with this manuscript. This does not alter our adherence to PLOS ONE policies on sharing data and materials.

by demonstrating how both approaches can be used in tandem to emphasize the positive aspects of each methodology while minimizing their respective drawbacks.

# 1. Introduction

## 1.1 The emergence of infodemiology in the early internet era

Since the 2016 US presidential election, online misinformation spread had become a ubiquitous topic in public discourse with rising concerns related to the proliferation of "fake news" on social media platforms such as Facebook and Twitter before the election [1–3]. Public concern about misinformation spread has only grown since then, with the World Health Organization (WHO) declaring an 'infodemic' in reaction to the proliferation of misinformation related to the COVID-19 pandemic [4]. However, risk related to online misinformation has existed since the early stages of the internet, as shown by the emergence of the field of infodemiology during that era [5, 6]. Infodemiology, also known as "information epidemiology," is the study of the determinants and distribution of health information and misinformation, and identifies areas where there is a knowledge translation gap between best available evidence and what most people do or believe, as well as markers for "high-quality" information [5, 7]. Infodemiological frameworks make distinctions between supply-based applications, such as analyzing what is being published on web sites and social media, and demand-based methods that examine search and navigation behavior on the internet [7]. Applications of infodemiology include the analysis of queries from internet search engines to predict disease outbreaks [6], monitoring peoples' health status updates on platforms such as Twitter and Weibo for syndromic surveillance [8–10], identifying prevalent themes and discourse around health conditions and behaviors (including COVID-19) [7, 11, 12], and studies examining online mobilization of social media users to influence health policy outcomes [13, 14].

While social media platforms are an incredible tool for staying connected and communicating with others, they can also simultaneously be a source of uncertainty and fear, which are often accompanied by increased dissemination of unverified rumors, misinformation, and fringe or conspiracy theories [15, 16]. To minimize the spread of misinformation, it is crucial to rapidly identify and then systematically sift through the high volume of posts and comments that often accompanies related online narratives, such as claims stating false ties between COVID-19 and 5G wireless technology. In response to this growing need for rapid content characterization, particularly in the context of health emergencies, supervised and unsupervised machine learning approaches, such as natural language processing (NLP) and supervised learning trained on fact-checked data, have been used to identify and classify misinformation [17–20]. Additionally, large language models (LLM) have shown high accuracy scores for identifying misinformation [21, 22], making popular models like chatGPT another promising tool for future infodemiological research. Despite the fact that various machine learning approaches are effective at classifying information and are crucial for adequately addressing the ongoing proliferation of misinformation that is overwhelming the social media ecosystem, these methods can be limited by their associated lexicon or dictionary and the semantic shifts that naturally occur in language over time and across contexts, including changing evidence about COVID-19 and other public health issues that may arise. Thus, for emerging issues that would not be recorded within training datasets, maintaining the use of human review and annotation remains necessary for properly detecting and contextualizing novel situations and narratives.

To gain a more thorough understanding of the nuanced patterns and dynamics underlying the spread of misinformation, our study introduces a hybrid analytic approach using metadata from 5G-COVID conspiracy discourse on Twitter (currently known as 'X' but referred to as Twitter for this paper) aimed at leveraging both the efficiency of NLP techniques and the qualitative schema afforded by human coders. More specifically, this study used topic modeling and sentiment analysis to identify influential posts and characterize the discourse, and then used both inductive and deductive coding to detect context-specific narratives that would not be measured by standard sentiment dictionaries. While NLP and qualitative coding methods have been widely used throughout the literature for identifying and characterizing misinformation (see the following for examples: [13, 23–25]), this paper combines these methods into a streamlined approach that can be utilized for rapidly characterizing nuanced themes and emerging narratives within large scale online discourses that requires significantly less burden on human annotation, as illustrated in the flow diagram in **S1 Appendix**. The hybrid method utilized in this study also has the potential to generate nuanced training data for machine learning classifier models without the need for annotating thousands of posts.

This study aims to further bridge the gap between computational and qualitative methodologies by demonstrating how NLP and manual annotation can be used synergistically to emphasize the positive aspects of each approach while minimizing their respective drawbacks. In order to assess the utility of this methodology, themes generated by this approach will be compared for consistency with 5G-COVID conspiracy themes previously identified on Twitter [26–28], Facebook [29], and Instagram [30]. Further, this study will extend the current literature by providing in-depth characterization of discourse focusing on correcting false information, which can be used to inform counterstrategies for online misinformation propagation.

## 1.2 Using machine learning to detect misinformation

Even before the proliferation of misinformation spurred by the emergence of COVID-19, there have been several research efforts over the past decade demonstrating that machine learning approaches can be effective methods for detecting misinformation. Previous work has developed models with at least 70% accuracy for correctly classifying misinformation across multiple platforms and topics, including Twitter posts related to the zika virus [31, 32], YouTube videos about prostate cancer [33], and false health-related information on medical forum posts [34] and Chinese posts on the platforms Weibo and WeChat [35]. Social media users can be classified as well, as shown by Ghenai and Mejova [36], who were able to identify with a 90% accuracy Twitter users prone to sharing false information promoting ineffective cancer treatments.

Many investigations using misinformation classifiers also examine the role of feature selection (i.e., what variables to include in the model) on prediction accuracy. While classifiers can effectively detect misinformation from only using textual features of the post, which are typically keywords associated with specific conspiracy theories or false narratives [37, 38], the inclusion of variables accounting for properties of the post beyond the content of the specific message, such as emotional sentiment, has been shown to improve model performance across multiple studies [17, 18, 20]. Deriving sentiment features from posts only requires basic arithmetic, where sentiment scores are calculated by first counting the number of words associated with a sentiment category and then dividing that count by the total number of words within the post [39]. Each sentiment category includes a dictionary, which is the list of words selected to represent the category. Sentiment dictionaries can measure a wide array of topics, such as emotional affect states (e.g., anger, anxiety, joy), biological processes (e.g., health, death), cognitive processes (e.g., certainty, causation), time orientations (e.g., focus on past, present, or future), and linguistic properties (e.g., number of 1st-person pronouns used) [40].

Assigning words to sentiment dictionaries can be accomplished using data-driven approaches, as shown by Stanford's Empath project that derived over 200 classification categories from analyzing more than 1.8 billion words of modern fiction [41]. However, for online misinformation research, psychometrically validated sentiment dictionaries provided by the software Linguistic Inquiry and Word Count (LIWC) have been widely used for characterizing misinformation discourse [42, 43] and calculating sentiment features across multiple studies that evaluate performance of misinformation classifiers [44–46]. A likely reason sentiment features are effective predictor variables is due to the fact that texts containing misinformation and conspiracies often have an emotional signature such as higher frequency of anger and anxiety words [15, 16, 43], although recent work indicates that emotional valence can vary depending on the type of misinformation being studied [47].

Despite the demonstrated utility of machine learning approaches for static misinformation detection, sentiment analysis and misinformation classifiers are both limited in their ability to adapt to the ever-evolving nature of human language. That is, in the same way that Gestalt psychology demonstrates that the perception of a color shifts depending on what other colors surround it [48], the shades of meanings from words also shift based on the other words surrounding it within a sentence or paragraph. Since word meanings can change due to differences in context-use and language shifts throughout time, results from sentiment scores and classifier models may not always reflect the actual semantic meaning of texts. These limitations can be clearly depicted in other applications such as automated detection models for abusive language as demonstrated by Yin and Zubiaga [49], who show that these classifier models are limited in their generalizability for other abusive language datasets, and may over-rely on the appearance of keywords such as slurs and profanity. While slurs and profanity can be strong predictors of online abusive language, abuse can also be expressed using implicit meanings and subtext, which results in models that overlook abuse in posts not including slurs and profanity. It is also possible that a post containing profanity keywords is not abuse at all, such as instances of teasing between friends, yet a model would falsely label as abuse [49]. Due to the continually evolving nature of language and context-dependent meanings, detecting misinformation and conspiracies only using machine learning models faces similar semantic challenges.

Interpreting sentiment scores can be further complicated when accounting for the fact that the discussion topic can also influence the average emotional tone of a discourse. For example, discussions about a pandemic may have higher percentages of negative affect words (e.g., "death", "tragedy") compared to lighter conversation topics such as gardening. Hence, it is difficult to determine what is an appropriate threshold for meaningful sentiment scores across topics. While this limits what researchers can infer from these analyses, this study addresses this limitation by using a relativistic interpretation of sentiment scores. More specifically, discourse will be marked as high in a sentiment category based on whether it is in the 90th percentile of scores within the corpus. Using the 90th percentile as a threshold marker allows us to account for the specific context of the 5G discourse when making judgements for determining what comprises a high level of sentiment. This approach is also adaptable across a wide array of topics since percentiles indicate which cases are high or low for a given metric based on the sample distribution. To further illustrate this point, a post containing 5% of death-related words may be in the 95th percentile for discourse about gardening, making it a "high" amount, but be within the 50th percentile for pandemic-related discourse, making it a typical percentage within the context of that corpus. Additionally, this study incorporates a qualitative coding approach to account for contextual information that is typically overlooked in sentiment analysis but recognizable by a human coder.

## 1.3 Inductive and deductive approaches for content coding

In recent years there have been calls for researchers to recognize the impact of contextualization when interpreting data, such as accounting for changes in semantics within a dataset over time [50]. When identifying emerging narratives within online discourse, where meanings of text vary due to novel co-occurrences of words, the boundaries between noise and relevant signal are constantly shifting. Compared to supervised machine learning approaches, which requires an existing training dataset of annotated posts to be effective, human coders are more adaptable to updating their background knowledge of events, making them effective at recognizing text containing previously undocumented narratives. To account for contextualization factors, our study utilized both inductive and deductive analytic techniques to examine tweets related to conspiracy discourse stating a relationship between COVID-19 and 5G technology.

Based in grounded theory [51], *inductive coding* is an iterative data analytic process centered on constant examination and comparison, which allows for theory development and explicit coding procedures [52, 53]. The primary benefits of an inductive approach is that it allows researchers to code texts using labels that are both aligned with the data and free from the influence of extant concepts, as well as detect tacit elements or connotations of the data that may not be apparent from a superficial reading of denotative content [54]. For the current study, inductive coding was used to identify narratives and rumors prominent within the 5G discourse.

*Deductive coding*, on the other hand, refers to a top-down coding process intended on testing whether data coincide with existing assumptions, theories, or hypotheses [55]. For the deductive coding scheme in the current study, we coded for whether posts contained misinformation or misinformation corrections based on whether it made statements claiming that 5G wireless technology causes COVID-19, or explicitly refuted the conspiracy. The criteria for classifying 5G-related misinformation and corrections were adapted from previous frameworks identifying COVID-related misinformation [56–58]. The researchers also coded for whether tweets expressed a positive, negative, or neutral stance towards 5G conspiracies. Coding for the user's stance makes it possible to trace and compare general sentiment across topics without having to account for specific themes.

By using both inductive and deductive coding schemes in our content analysis, we aimed to extract and interpret the underlying patterns of COVID-19 misinformation in a more comprehensive manner that are both consistent with the textual data and aligned with existing conceptual frameworks. This approach also allows us to track both general conceptual categories (e.g., misinformation) and more case-specific incidents (e.g., religious pastor claiming 5G causes COVID-19), or in other words, seeing both "the forest and the trees."

## 1.4 Combining unsupervised machine learning and content analysis approaches

Previous studies that only use content analysis when investigating social media activity can require the coding of hundreds, if not thousands of posts, which can be an extensive cost in time and resources. Fortunately, unsupervised machine learning and NLP approaches have been used to aid in content analysis on social media, which includes the detection of illicit drug sales or promotions [59–62], self-report symptoms on social media [63], and identifying prominent themes in COVID-related misinformation [56]. These approaches group posts together based on textual similarity, which can help filter out irrelevant topic clusters based on keywords [64], or organize the posts together to facilitate the identification of higher-level themes [56]. Since most social media discourse is driven by a small number of highly active users while the majority of users typically engage in passive behaviors such as browsing [65–

68], characterizing discourse based on the most highly shared posts can be an effective approach for assessing public response to an issue due to most users not often generating their own content. This is further demonstrated in a recent study that conducted a social network analysis of Twitter users, which showed that the propagation of a COVID-related conspiracy theory was initiated by only a handful of prominent accounts during the early stages of the pandemic [69].

Another advantage of applying topic modeling to social media data is that it allows for other prominent sub-themes to be identified from the topic clusters that might have been over-looked by just assessing a more general sampling of the most shared posts, such as the most retweeted tweets. This is demonstrated in previous work by the authors [56] who were able to identify themes related to the misuse of scientific authority within Twitter misinformation discourse after applying biterm topic modeling (BTM) to the data and then coding the top 10 retweeted tweets from each topic cluster. Using this approach, informally referred to as "BTM + 10," we were able to build on themes identified in their previous characterization of misinformation discourse based on a more general sampling of the top 100 retweeted tweets [58]. While BTM + 10 does not cover the content in every post assigned to a topic, the top 10 most retweeted tweets were able to account for at least 50% of the total tweet volume of each topic cluster within the context of the hydroxychloroquine twitter discourse [56], indicating that influential posts tend to account for a majority volume of tweets within a discourse. It is also worth noting that uncoded tweets are still assigned to a topic cluster based on textual similarity, which increases the likelihood that it would be similar in message content to the most influential posts that were assigned to the same cluster. Increasing the number of coded posts from the 10 most retweeted tweets to 15 or 20 can also account for greater tweet volume without adding substantial burden on the coders. In the current study, we wish to build on the BTM + 10 methodology as carried out in previous work [56] by incorporating sentiment analysis when characterizing 5G-COVID conspiracy discourse on Twitter. Themes identified from previous studies examining 5G conspiracy discourse will be compared to assess the validity of the methodology proposed in the current study. For readers interested in adapting this methodology, see the flow chart diagram in **S1 Appendix** that further outlines the current approach.

## 1.5 5G conspiracy theories during the COVID-19 pandemic

Beginning in early April 2020, there had been reports that telecom engineers were facing verbal and physical threats, and that at least 80 mobile towers had been burned down in the United Kingdom (UK), actions fueled by false conspiracy theories blaming the spread of COVID-19 on 5G wireless signals [70]. In order to increase understanding of these destructive acts, recent work has shown associations between belief in 5G-COVID conspiracies with states of anger and greater justification of real-life and hypothetical violence alongside greater intent to engage in similar behaviors in the future [71].

Public figures and celebrities who, whether with deliberate malintent or not, share rumors and falsehoods to their large groups of followers [72–75] are also involved in 5G-COVID conspiracy discourse, as shown in a recent study characterizing discourse on Facebook that identifies celebrities and religious leaders as 5G-COVID conspiracy propagators [29]. Bruns et al. (2020) also identify prominent 5G conspiracy theories such as claims stating that 5G reduces the ability of the human body to absorb oxygen, and that 5G is related to a complex agenda involving bioengineered viruses and deadly 5G-activated vaccines led by elite figures such as Bill Gates, George Soros, the World Health Organization (WHO), and secret-society organizations like the Illuminati [29]. From these findings, Bruns et al. (2020) conclude that 5G

conspiracy theorists had retrofitted the new information emerging about the virus and its effects on human health into pre-existing worldviews, beliefs, and ideologies to further propagate conspiracist narratives [29]. This is consistent with findings from another 5G conspiracy study conducted on Twitter, which shows that the conspiracies were built on existing ideas set against wireless technologies [28]. On Twitter more specifically, researchers found that videos played a more crucial role in 5G rumor propagation than posts [28], and other work has examined how spatial data has been misconstrued by conspiracists, as shown with the promotion of maps that assert false correlations between the distribution of COVID-19 cases and installations of 5G towers [27].

Another study that used social network analysis found that influential accounts tweeting 5G-COVID conspiracies tended to form a broadcast network structure resembling the structure most typical for accounts from mainstream news outlets and celebrities that are frequently retweeted [26]. A content analysis from this study also shows that over a third of randomly sampled tweets contained views claiming that COVID and 5G were linked, and that there was a lack of an authority figure who was actively combating said misinformation [26]. In order to examine the authors of influential tweets within 5G discourse, the current study will also content code affiliations from the most retweeted user accounts based on publicly available profile data.

## 2. Methods

### 2.1 Data collection and analysis overview

A total of 256,562 tweets were collected from the public streaming Twitter API per the terms available at the time of the study using keywords "5G" and covid-related words such as "coronavirus", "covid-19" between March 25th and April 3rd 2020. We chose this time frame as it represents a period when the 5G conspiracy theory first became prominent, as shown in the spike in volume for "5G" posts in Fig 1. All personal identifiable information from tweets was removed in the reporting of the results to preserve anonymity. We note that due to the change

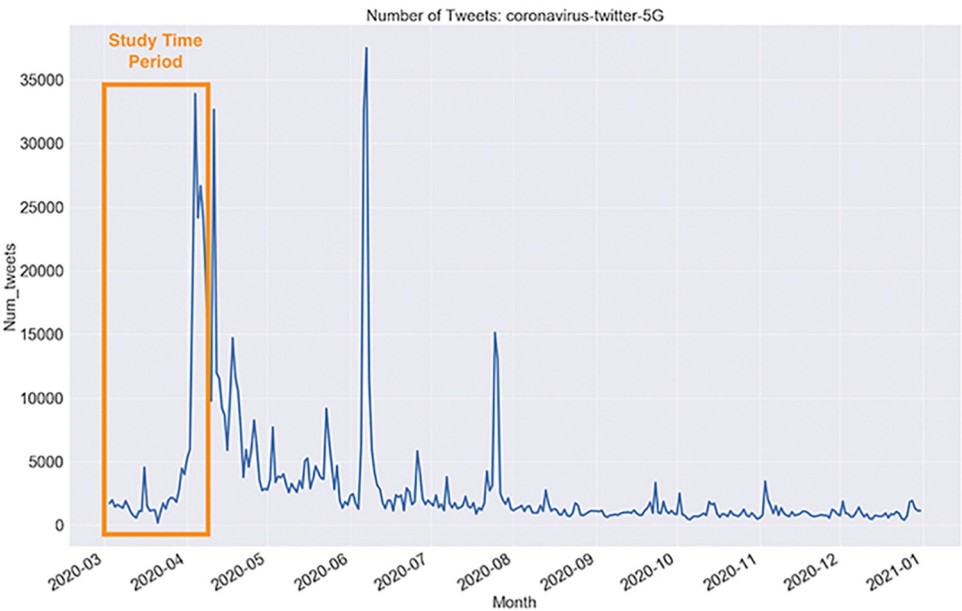

**Fig 1. Number of Tweets with keyword "5G" through time highlighting timeframe of current study.**

in ownership, API policies, and name of the platform (Twitter has been renamed "X"), the terms and conditions of the streaming API used for data collection for this study are no longer applicable for current studies. IRB approval was not required as all data collected in this study was available in the public domain and results from the study have been deidentified and anonymized. The dataset and R syntax used to generate the results can be found at the following Open Science Framework (OSF) link: https://bit.ly/5G_Conspiracies.

To analyze the relatively large volume of tweets collected in this study, we used the biterm topic model (BTM), an unsupervised machine learning approach using natural language processing (NLP) further described in the next section, to extract themes from text of tweets as used in prior studies examining COVID-19 topics on social media [10, 56, 63, 64]. The top 10 most retweeted tweets associated with each topic cluster (i.e., the "BTM + 10" approach previously mentioned) were coded using a deductive coding scheme adapted from previous COVID-19 misinformation work [56, 58] to classify posts on whether they contain misinformation, or a misinformation correction (further discussed in Section 2.3). For tweets that may not contain misinformation but still support the notion that 5G signals cause COVID-19 infections, positive and negative stances were coded to assess whether a tweet supported or opposed the conspiracy (Table 1). While the metric of stance is traditionally referred to as "sentiment" in related social media research, this current study will refer to it as "stance" to prevent potential confusion with NLP sentiment analysis metrics further described in the next section. An inductive coding scheme was also used to characterize themes and narratives for both misinformation and correction discourses.

## 2.2 Biterm Topic Model (BTM)

Unsupervised topic modeling strategies, such as BTM, are methods particularly well suited for sorting short text (such as the 280-character limit for tweets) into highly prevalent themes without the need for predetermined coding or a training/labelled dataset to classify specific content. This is particularly useful in characterizing large volumes of unstructured data where predefined themes are unavailable, such as in the case of emerging social movements, novel

Table 1. Examples of content coded tweets (paraphrased and redacted to retain anonymity).

| *Misinformation Categories* | *Misinformation*:<br>1. #5G produces the same symptoms as this supposed #coronavirus or #COVID19. Watch this video [LINK]<br>2. RT! 5G is the real silent killer, not the Corona Virus!!! |
|---|---|
| | *Misinformation Correction*:<br>1. Scientists say any suggestion that coronavirus and 5G are linked is "complete rubbish" and biologically impossible<br>2. I can confirm that 5G is in no way giving people #coronavirus because that would require converting radio waves into organic molecules. This is pretty much God-level ability, and anyone capable of it would be running the planet already |
| *Stance towards 5G Conspiracy* | *Positive Stance*:<br>1. #Ireland. You're about to be fried. We have pleaded with people to listen to the 23,000 scientists who want #5G banned<br>2. Unnamed 5G #Whistleblower Claims That People Are Being Infected With #Coronavirus Via #Covid19 Tests |
| | *Negative Stance*:<br>1. Imagine being months into a relationship with a person you could see a potential future with, and one morning they send you links and paragraphs on the 5G conspiracy being the source of the coronavirus.<br>2. If you're into this 5G conspiracy theory, here are some other just as likely causes of Covid-19 to look into: ferns, Ed Sheeran, clams and clam sauce, those disposable flossers, Golden Retrievers & hemp granola. |

disease outbreaks, and other emergency events where information changes rapidly [10, 56, 60, 61, 63, 64, 76, 77]. The corpus of tweets containing the 5G keywords was categorized into highly correlated topic clusters using BTM based on splitting all text into a bag of words and then producing a discrete probability distribution for all words for each theme that places a larger weight on words that are most representative of a given theme [78]. While other NLP approaches use unigrams or bigrams for splitting text, BTM uses 'biterms', which is a combination of two words from a text (e.g., the text "go to school" has three biterms: 'go to', 'go school', 'to school') and models the generation of biterms in a collection rather than documents [79].

BTM was used for this study because biterms directly model the co-occurrence of words, which increases performance for sparse-text documents such as tweets. Conducting BTM analysis is done initially by setting the BTM topic number (k) and "n" words (for the first round of analysis we set at k = 10, n = 20 to cover several possible misinformation topics that might be present in the corpus). A coherence score is then used to measure how strong the top words from each topic correspond to its respective topic. For this study, the model with k = 20 was chosen because it had the highest coherence score compared to other iterations tested. All data collection and processing were conducted using the programming language Python.

## 2.3 Deductive and inductive coding schemes

In order to characterize highly prevalent misinformation and conspiratorial narratives in the corpus, the top 10 most retweeted tweets from all 20 BTM topic outputs were extracted and manually coded for relevance first using a deductive coding scheme adapted from existing COVID-19 misinformation themes from the literature [56–58], and then coded again using an inductive approach identifying context specific themes related to 5G. While misinformation and conspiracies are distinct concepts, the current study will refer to both as 'misinformation' within the analysis for brevity.

In total, 200 unique tweets were reviewed by coders. Tweets were classified as misinformation if they contained declarative statements claiming that 5G causes COVID-19, or statements supporting the conspiracy from sources that convey scientific authority such as from medical experts, scientists, or scientific studies [56]. A tweet was considered a misinformation correction if it explicitly opposes the 5G conspiracy and provided information countering the claims about 5G causing COVID-19. Tweets were also annotated for stance in relation to 5G conspiracy theories, with 1 indicating positive stance, -1 indicating negative stance, and 0 if the tweet only reports information about 5G without stating an opinion, exhibits neutral user sentiment, or is not directly related to the 5G COVID conspiracy discourse.

An inductive coding approach was then used to sub-code for reoccurring themes and narratives associated with each topic cluster that is unique to 5G-related misinformation (see Table 2 below for a description of each identified sub-theme and example tweets). Of the total 200 tweets used for inductive coding, the first and third author divided the sample in half to identify themes. Once theme labels were generated by each coder, both coders met to compare inductive coding labels and combine overlapping themes. All 200 tweets were then reviewed again to classify them based on the finalized coding scheme. Using this approach, tweets can be categorized with the same theme label from inductive coding but classified as misinformation or a correction based on the deductive framework. For example, a tweet labeled as misinformation and the inductive coding theme *causative explanation* (see Table 2 for description) indicates that the tweet is making a causal claim about a correlation between COVID-19 and 5G. However, a tweet labeled as a misinformation correction but labeled as the same theme indicates that the tweet is making causal claims refuting correlations between COVID-19 and

**Table 2. Inductive coding themes.**

| Theme | Classification Label | Description | Example Tweet (paraphrased to preserve anonymity) |
|---|---|---|---|
| Causative explanations of 5G & COVID | Caus Exp | Claims about the causal effects of 5G and COVID-19, sometimes using scientific language (e.g., describing cause and effect mechanisms) | "[Redacted] declares that COVID-19 is not a virus at all, but a reaction to the radiation created by 5G technology." |
| Medical authority | Med Aut | Claims made from or endorsed by medical authority figures (such as medical scientist, doctor, or scientific study) | "Dr. [Redacted], M.D. hypothesizes that Coronavirus may be history repeating itself & caused by 5G." |
| Religious figure | Pastor | Messages made by religious pastors about the connection between 5G and COVID, or messages in response to the pastor's remarks | "Pastor [Redacted] said corona virus doesn't kill but 5G does. He said telecom companies are installing 5G networks in lagos and Abuja that's why federal government enforced a lockdown on both states" |
| Celebrity | Celeb | Statements addressing celebrities who state their stance on 5G-COVID conspiracy theories | "[Celebrity Singer] out here spreading conspiracy theories that corona virus is a result of 5G networks" |
| Antivax | Anti-Vax | Claims that vaccines, including future COVID-19 vaccines are linked to conspiracy | "The entire Pandemic crisis has 3 components. . . First Vaccines to increase metal levels in body. Then bombard the body with high energy radiation to destroy the immune system. Then launch the virus to create the Pandamic. Check the date of 5G launch in Wuhan. . ." |
| Negative health effects | Neg Health | Statements claiming exposure to 5G causes negative health effects | "5G just rolled out in Europe, China, and North America. Microwave radiation cripples immunity and increases inflammation. It's the perfect fuel for a pandemic. We have to stop it. 5G: Fueling COVID" |
| Video | Video | References to video being shared on YouTube about 5G conspiracies | "Corona = 5G weapon used as disguise of virus? This video is VERY shadow banned. It won't show up to my history.. . ." |
| Technology correlations with disease outbreaks | Tech Corr | Statements with timelines showing disease outbreaks alongside the release of new radio wave technologies | "2003 - 3G introduced to the world<br>2003—SARS outbreak<br>2009 - 4G introduced to the world<br>2009—Swine flu outbreak<br>2020 - 5G introduced to the world<br>2020—Coronavirus outbreak" |
| Huawei | Huawei | Story about the UK government pulling out of 5G contract with Chinese communications company Huawei | "Britain pulls out of 5G contract with Chinese firm Huawei after test kits were found contaminated with Corona virus. . ." |
| Group of elites | Elite | Statements referring to group of elites, which typically included figures such as Bill Gates, Elon Musk, and the illuminati | "If people knew what was unfolding & what satan &his Illuminati agents, his disciples like @elonmusk & @BillGates & the proponents of the New World Order have in mind &are doing with 5G, coronavirus & the coming Anti-coronavirus vaccines, everyone would join forces & PRAY!" |
| Towers on fire | Tower Fire | Discussions about people setting 5G towers on fire | "UK mobile phone masts torched and engineers abused over "baseless" theories linking coronavirus to 5G" |
| Geography comparisons | Geo | Posts comparing maps of 5G coverage and COVID outbreaks, or mention areas that have 5G coverage and no COVID cases or vice-versa | "Let's take a quick look at Italy. Left: Coronavirus cases Right: Number of 5G deployments with commercial available tech" |

5G technology. An advantage of this approach is that it showcases overlap in discourse themes between misinformation and correction topic clusters.

## 2.4 Content coding of Twitter account profiles

Twitter profiles of accounts that produced the top 10 most retweeted tweets for each BTM cluster output were content coded to investigate publicly self-reported occupation among these influential users who were active in the Twitter 5G conspiracy discourse during the study period. Publicly available metadata from user account profiles were retrieved and coded to determine whether descriptions stated that they were a medical doctor or scientist, a religious

leader, a government official, or affiliated with the media (e.g., a journalist, TV, or radio personality).

## 2.5 Calculating LIWC sentiment scores

Sentiment scores were calculated using Linguistic Inquiry and Word Count (LIWC) and reflect the percentage of words within a post that correspond to a given sentiment category. The sentiment scores were calculated to assess language related to: Analytic thinking (metric of logical, formal thinking), Clout (language of leadership, status), Authenticity (perceived honesty, genuineness), and Netspeak (internet slang), cognitive-processes related to information evaluation (i.e., Causation, Discrepancy, Tentative, Certitude), Emotional Affect (i.e., Affect, Positive Tone, Negative Tone, Emotion, Positive emotion, Negative emotion, Anxiousness, Anger, Sadness), social- and health-related topics (i.e., Prosocial behavior, Interpersonal conflict, Moralization, Health, Illness, Death, and Risk), and time-related sentiments (i.e., Time, Past focus, Present focus, Future focus). Since some tweets receive a much greater proportion of engagement within a discourse than others, average sentiment scores for each topic cluster were also weighted based on number of retweets received. This allowed us to characterize emotional sentiment of a topic cluster based on the most influential posts, rather than by posts that received low exposure. Due to the lack of a normal distribution of sentiment scores, we applied a Kruskal-Wallis nonparametric approach and Dunn's post-hoc test to detect statistically significant differences across BTM topic clusters, with a Bonferroni Correction to account for multiple comparisons.

## 3 Results

### 3.1 Content analysis and characterization

From the 256,562 tweets analyzed in this study, Table 3 shows all BTM topics generated and assessed. In this study, the majority of topics have at least 50% of their tweet volume composed of the top 10 tweets with highest volume of retweets, indicating that the top 10 retweets characterize a substantive volume of tweets assigned to each topic. There were 4 topics below 50%, with the lowest percentage being 40.9%. While this still accounts for a substantive percentage of tweet volume, it should be noted that there is more uncertainty associated with the characterizations based on the metrics derived from the top 10 most retweeted tweets for these topic clusters. Topics were further classified as high in misinformation or high in misinformation corrections if the topic had at least 33% of top 10 retweeted tweets associated with the respective categories. Topics were also labeled high in both (i.e., Mixed) if it contained at least 33% of misinformation and corrections, and low for the remaining topics. A threshold of 33% was chosen for this analysis since it indicates a substantive number of tweets for each topic. Out of the 20 topics in this study, 9 were classified as high in misinformation, 5 as high in misinformation corrections, 2 as high in both misinformation and corrections, and 4 as low in both. The topic clusters low in both were not chosen for further analysis.

Table 4 shows each topic cluster with the themes identified from inductive coding, as well as a list of LIWC sentiment categories in which the topic scores in the 90th percentile. Most of the topic clusters score in the 90th percentile for at least one sentiment category, indicating that the topic clusters cover a wide range of emotional sentiments. The results in Table 4 show that themes were spread across topics, with sentiment varying across the discourse as well. While some topic clusters are predominately associated with one theme, as seen in topics 7, 8, 11, and 12, other topic clusters have multiple themes. The misinformation topic with the highest number of themes is topic 10, which includes themes of *causative explanations*, *medical authorities*, *negative health effects*, and *video*. Within the correction discourse, *medical authority* was paired

**Table 3. Properties of BTM clusters.**

| TOPIC CLUSTER ID | BASED ON ALL TWEETS PER CLUSTER | | | | | BASED ON TOP 10 RETWEETED TWEETS BY CLUSTER | | | |
|---|---|---|---|---|---|---|---|---|---|
| | Total Tweets | Unique tweets | Top10% | Twitter URL | YouTube URL | Avg Stance | Stance Level | Misinfo % | MisCorr % |
| 0 | 5669 | 1617 | 58.2% | 48.6% | 16.6% | 0.766 | Strongly Positive | 94.0% | 0.0% |
| 1 | 14122 | 3529 | 54.7% | 77.6% | 1.2% | -0.166 | Mixed | 15.1% | 43.8% |
| 2 | 66434 | 15967 | 71.9% | 48.4% | 1.7% | -0.035 | Mixed | 11.2% | 1.1% |
| 3 | 14857 | 4490 | 48.7% | 33.8% | 1.2% | -0.793 | Strongly Negative | 0.0% | 80.3% |
| 4 | 9703 | 1988 | 57.2% | 76.5% | 4.8% | 0.755 | Strongly Positive | 87.8% | 12.2% |
| 5 | 5939 | 2023 | 63.6% | 61.9% | 1.4% | -0.345 | Negative | 59.6% | 35.1% |
| 6 | 2217 | 703 | 80.5% | 46.9% | 0.1% | 0.118 | Mixed | 43.6% | 0.0% |
| 7 | 1472 | 1358 | 50.3% | 49.6% | 35.6% | 1 | Strongly Positive | 100.0% | 0.0% |
| 8 | 2429 | 892 | 72.7% | 69.5% | 9.8% | 0.81 | Strongly Positive | 69.9% | 3.1% |
| 9 | 7030 | 3219 | 46.6% | 71.4% | 0.5% | -0.296 | Negative | 31.7% | 52.2% |
| 10 | 2551 | 727 | 80.7% | 84.7% | 3.7% | 0.606 | Positive | 60.6% | 0.0% |
| 11 | 2358 | 345 | 89.1% | 79.4% | 0.6% | 0.926 | Strongly Positive | 92.6% | 0.0% |
| 12 | 20520 | 3594 | 47.0% | 76.4% | 2.4% | 0.448 | Positive | 12.4% | 0.0% |
| 13 | 10478 | 700 | 88.1% | 73.7% | 1.1% | 0.97 | Strongly Positive | 92.7% | 0.0% |
| 14 | 4765 | 1510 | 61.2% | 78.0% | 4.2% | 0.46 | Positive | 27.9% | 3.2% |
| 15 | 5866 | 770 | 86.5% | 68.7% | 10.9% | 0.162 | Mixed | 17.2% | 5.7% |
| 16 | 1523 | 628 | 68.0% | 56.5% | 22.3% | 1 | Strongly Positive | 93.3% | 0.0% |
| 17 | 18512 | 4788 | 54.2% | 31.9% | 2.3% | -0.871 | Strongly Negative | 0.0% | 83.8% |
| 18 | 50014 | 19149 | 40.9% | 61.5% | 1.3% | -0.847 | Strongly Negative | 4.4% | 65.2% |
| 19 | 10103 | 2625 | 76.3% | 37.3% | 2.9% | -0.098 | Mixed | 41.2% | 40.4% |

Columns "Misinfo %" and "MisCorr %" indicate the percentage of tweets categorized as misinformation or misinformation corrections respectively and weighted by number of retweets. The column "Avg Stance" is based on the average stance scores of the top 10 retweeted tweets for each topic. Additionally, the column "Top10%" indicates the percentage of total tweets for each topic that is accounted for by the top 10 retweeted tweets.

with themes related to *5G towers on fire* and *causative explanations* in topics 3 and 18. Among the topic clusters that were predominately misinformation, none of the correction tweets corresponded to any of the inductive coding themes. This suggests that misinformation discourses are closed off and less likely to have involvement from those providing corrections. However, this is not consistent with topic clusters high in corrections, as shown in topic 1 where corrections that include *causative explanations* are in the same discourse as *anti-vax* misinformation tweets, and topic 9, which has both misinformation and correction tweets that mention *religious figures*. These results could indicate misinformation discourse is more likely to interfere with correction discourse. The correction discourse also does not mention themes for *Tech Corr* (i.e., claims that wireless technology advancements are correlated with disease outbreaks), *Elite* (i.e., claims that figures like Bill Gates and Elon Musk are promoting COVID-19 with 5G technology), *Huawei* (i.e., the UK government pulling out a contract from a Chinese communications company due to health concerns over 5G), *anti-vax* (i.e., claims about future COVID-19 vaccines having adverse health effects) and *Geo* (i.e., maps showing overlapping distribution between COVID cases and 5G cellular towers), indicating that the corrections are not addressing specific misinformation narratives.

Table 5 groups each topic cluster together based on misinformation classification and shows the percentage of coded tweets corresponding to each inductive coding theme. Among misinformation topics, the themes *Huawei* (33.5%), *video* (14.1%), and *anti-vax* (9.9%) were the three most prominent. The three most prevalent themes for corrections were *medical authority* (22.1%), *causative explanations* (10.3%), and *celebrities* (10.1%). The difference in

**Table 4. Inductive coding themes and LIWC sentiment scores across topics.**

| | TOPIC CLUSTER ID | MISINFO THEME (%) | CORRECT THEME (%) | LIWC SENTIMENT |
|---|---|---|---|---|
| **HIGH IN MISINFORMAITON** | 0 | **Video** (94.6%) **Neg Health** (5.4%) | *None* | Netspeak Negative Tone Emotion Negative Emotion |
| | 4 | **Anti-Vax** (34.5%) **Elite** (29.5%) | *None* | Clout Moral Future Focus |
| | 6 | **Video** (37.1%) | *None* | Sadness Conflict Risk |
| | 7 | **Video** (91.8%) | *None* | Time Past Focused |
| | 8 | **Anti-Vax** (71.2%) | *None* | Health Illness |
| | 10 | **Caus Exp** (12.0%) **Med Aut** (4.6%) **Neg Health** (7.0%) **Video** (2.6%) **Tech Corr** (3.1%) | *None* | Discrepancy Health Death |
| | 11 | **Tech Corr** (100.0%) | *None* | Analytic Illness |
| | 13 | **Huawei** (98.1%) | *None* | Analytic Anger Past Focused |
| | 16 | **Caus Exp** (78.4%) **Med Aut** (21.6%) | *None* | Anxiety Pro-social |
| **HIGH IN CORRECTIONS** | 1 | **Anti-Vax** (26.6%) **Neg Health** (49.7%) | **Caus Exp** (86.1%) | Causation Tentativeness Present Focused |
| | 3 | *None* | **Med Aut** (67.3%) **Tower Fire** (32.7%) | *None* |
| | 9 | **Pastor** (30.9%) | **Pastor** (76.9%) | Clout |
| | 17 | *None* | **Celeb** (60.8%) | Negative Tone Emotion Negative Emotion |
| | 18 | *None* | **Caus Exp** (17.3%) **Med Aut** (54.5%) | Certitude Anger |

(*Continued*)

**Table 4.** (Continued)

|  | TOPIC CLUSTER ID | MISINFO THEME (%) | CORRECT THEME (%) | LIWC SENTIMENT |
|---|---|---|---|---|
| **MIXED** | 5 | **Geo** (50.5%) | *None* | Authentic Sadness Time |
|  | 19 | **Video** (24.9%) **Med Aut** (75.1%) | **Pastor** (77.1%) **Celeb** (22.9%) | Positive Tone Positive Emotion Moral Risk |

The column "Misinfo Theme (%)" lists the themes that appear in each topic with the percentage indicating the proportion of coded misinformation tweets that were also assigned the theme label. The column "Correct theme (%)" is similar to the previous column, except that it shows the proportion of coded correction tweets assigned to a theme.

prevalent themes further illustrates that the two conversations focused on differing conversation topics. Within misinformation discourse, the conspiracy theme *Huawei* claims that the UK government pulled out of a 5G contract due to concerns about it causing COVID, while the actual circumstance was due to concerns about surveillance from the Chinese government. In this case, a situation with a kernel of truth was misconstrued to perpetuate a false claim. The other themes *video* and *anti-vax* indicate conversations occurring off Twitter and plant the seed for future distrust in medical efforts. While the correction discourse made general refutations against the false correlation between 5G and COVID-19, it mostly did not address the specific narratives that were prominent in misinformation discourse, which could mitigate the effectiveness of the corrections. For mixed topic clusters, *religious figures* (29.8%), *medical authority* (20.8%), and *geographic comparisons* (9.9%) were the most common themes. It is possible that themes such as *religious figures* could be associated with conflicts between people's faith and public health guidelines, which may contribute to the high volume of both misinformation and corrections. It is also possible that the appearance of medical authorities both supporting and refuting claims that 5G causes COVID-19 could lead to ambiguity and confusion among users.

### 3.2 LIWC sentiment scores

LIWC sentiment scores were also averaged across BTM topics based on misinformation classifications as shown in **Table 6** and visualized as bar graphs in **Figs 2–6**. Topics with low amounts of misinformation and corrections were kept in this analysis for comparison. Sentiment categories listed in **Table 6** that had statistically significant differences across all misinformation classifications are marked with an asterisk (*). Among Text Analytic sentiments (i.e., *Analytic, Clout, Authentic,* and *Netspeak*), misinformation topics on average scored higher than corrections across all categories except *Authentic*, with these differences being statistically significant ($p < .001$). These differences are most pronounced when comparing *Analytic* (78.74% misinformation vs 67.66% correction) and *Clout* sentiments (46.57% vs 41.45%), indicating that misinformation topics were more likely to use words reflecting logic and social

**Table 5. Themes from inductive coding by misinformation classification.**

| TOPIC | CAUS EXP | MED AUT | PASTOR | CELEB | ANTI-VAX | NEG HEALTH | VIDEO | TECH CORRE | HUAWEI | ELITE | TOWER FIRE | GEO |
|---|---|---|---|---|---|---|---|---|---|---|---|---|
| **MISINFO** | 3.5% | 1.0% | 0.9% | 0.0% | 9.9% | 1.0% | 14.1% | 7.7% | 33.5% | 5.6% | 0.0% | 0.0% |
| **CORRECT** | 10.3% | 22.1% | 3.9% | 10.1% | 0.7% | 1.1% | 0.6% | 0.0% | 2.7% | 0.0% | 5.7% | 0.0% |
| **MIXED** | 0.0% | 20.8% | 29.8% | 6.2% | 0.0% | 0.0% | 6.9% | 0.0% | 0.0% | 0.0% | 0.0% | 9.9% |

**Table 6. Average LIWC sentiment scores (i.e., percentage of words corresponding to each sentiment category) across misinformation classifications.**

| Text Analytics | Topic | Misinformation | Correction | Mixed | Low |
|---|---|---|---|---|---|
| | Analytic* | 78.74 | 67.66 | 77.23 | 59.06 |
| | Clout* | 46.57 | 41.45 | 46.35 | 29.54 |
| | Authentic* | 20.49 | 21.34 | 18.24 | 27.06 |
| | Netspeak* | 5.10 | 4.71 | 4.64 | 6.44 |
| Cognitive | Topic | Misinformation | Correction | Mixed | Low |
| | Cause* | 0.78 | 2.06 | 1.58 | 1.54 |
| | Discrep* | 0.61 | 1.11 | 0.98 | 1.98 |
| | Tentat* | 0.75 | 1.66 | 1.30 | 2.54 |
| | Certitude | $0.07_{c,l}$ | $0.52_{mi,m,l}$ | $0.12_{c,l}$ | $2.02_{mi,c,m}$ |
| Affect | Topic | Misinformation | Correction | Mixed | Low |
| | Affect (general)* | 7.03 | 6.95 | 7.13 | 7.17 |
| | Positive Tone* | 0.67 | 0.82 | 1.30 | 0.81 |
| | Negative Tone* | 6.17 | 5.90 | 5.52 | 5.84 |
| | Emotion (general) | $5.26_{c,m}$ | $4.92_{mi,m,l}$ | $5.14_{mi,c,l}$ | $4.81_{c,m}$ |
| | Positive Emotion* | 0.06 | 0.14 | 0.40 | 0.23 |
| | Negative Emotion* | 5.15 | 4.69 | 4.66 | 4.54 |
| | Anxiousness* | 0.06 | 0.06 | 0.04 | 0.09 |
| | Anger | $0.08_{c,m,l}$ | $0.18_{mi,m,l}$ | $0.01_{mi,c}$ | $0.03_{mi,c}$ |
| | Sadness | $0.01_{c,m}$ | $0.02_{mi,l}$ | $0.02_{mi,l}$ | $0.01_{c,m}$ |
| Social/Health-related Topics | Topic | Misinformation | Correction | Mixed | Low |
| | Prosocial | $0.18_{c,m,l}$ | $0.29_{mi,l}$ | $0.25_{mi,l}$ | $0.22_{mi,c,m}$ |
| | Conflict* | 0.26 | 0.24 | 0.41 | 0.64 |
| | Moral* | 0.24 | 0.32 | 0.48 | 0.16 |
| | Health* | 4.80 | 4.62 | 4.63 | 4.61 |
| | Illness | $4.18_{c,m,l}$ | $4.21_{mi,l}$ | $4.08_{mi,l}$ | $4.35_{mi,c,m}$ |
| | Death* | 0.33 | 0.12 | 0.29 | 0.52 |
| | Risk* | 0.16 | 0.23 | 0.52 | 0.14 |
| Time | Topic | Misinformation | Correction | Mixed | Low |
| | Time (general)* | 2.66 | 1.78 | 2.65 | 1.45 |
| | Past Focus | $3.11_{c,m,l}$ | $1.51_{mi,l}$ | $1.59_{mi,l}$ | $1.46_{mi,c,m}$ |
| | Present Focus* | 3.59 | 4.98 | 4.75 | 7.16 |
| | Future Focus* | 0.53 | 0.58 | 0.47 | 0.53 |

Categories with

"*" indicate statistical significance (< .01) across all classifications using Bonferroni corrections to adjust for multiple comparisons. For categories without an "*", significance differences between classifications are depicted if the row contains the following subscripts: *mi* = Misinformation, *c* = Corrections, *m* = Mixed, and *l* = Low.

status compared to corrections. The mixed BTM topics that contained high levels of both misinformation and corrections scored second highest in *Analytic* (77.23%) and *Clout* (46.35%) sentiments but lowest in *Authentic* (18.24%) against all other classifications. In contrast, topics that contained low amounts of both misinformation and corrections on average scored the lowest in *Analytic* (59.06%) and *Clout* (29.54%), but highest in *Authentic* (27.06%) and *Netspeak* (6.44%).

For sentiments reflecting cognitive processes (i.e., *Causation*, *Discrepancies*, *Tentativeness*, and *Certitude*), corrections on average scored significantly higher than misinformation topics across all sentiment categories, and was highest in *Causation* words (2.06%) compared to all other topic classifications (p < .001). Topics low in both misinformation and corrections scored highest in *Discrepancy* (1.98%), *Tentative* (2.54%), and *Certitude* (2.02%) sentiments.

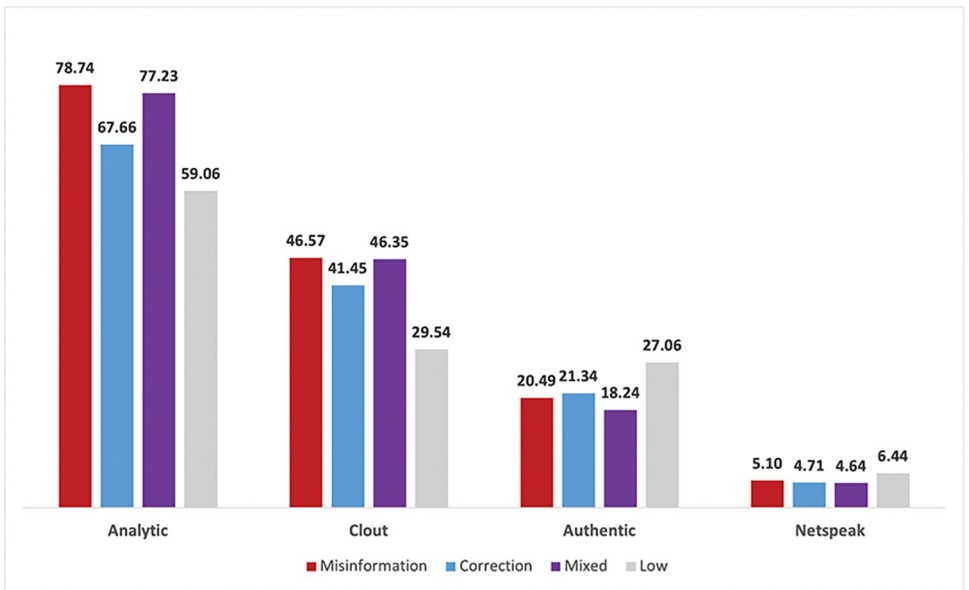

**Fig 2. Average percentage of text analytic sentiment across misinformation classifications.**

Among sentiment for general *Affect*, misinformation topics on average score significantly higher than corrections (7.03% vs 6.95%, p < .001), and are also higher in general *Emotion* (5.26% vs 4.92%, $p < .001$), *Negative Emotion* (5.15% vs 4.96%, $p < .001$), and *Negative Tone* (6.17% vs 5.90%, $p < .001$). However, misinformation scores lower than corrections in *Positive Tone* (0.67% vs 0.82%, $p < .001$), *Positive Emotion* (.06% vs .14%, $p < .001$), and *Anger* (0.08% vs 0.18%, $p < .001$). There are also statistical differences between misinformation and correction topics for *Anxiousness* (0.063% vs .058%, $p = .023$) and *Sadness* (0.01% vs 0.02%, $p < .001$), although the magnitude of the differences is fairly small.

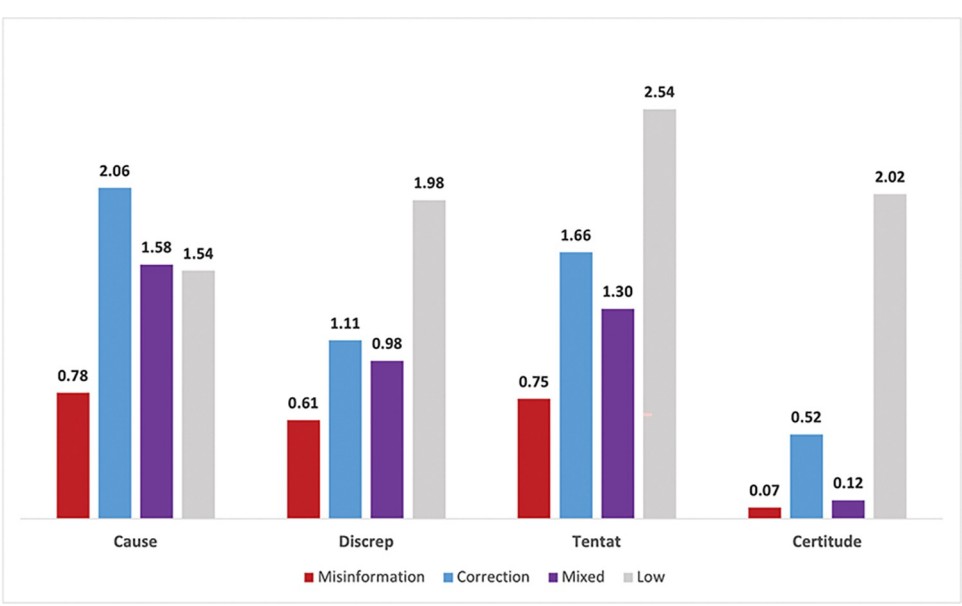

**Fig 3. Average percentage of cognitive sentiment across misinformation classifications.**

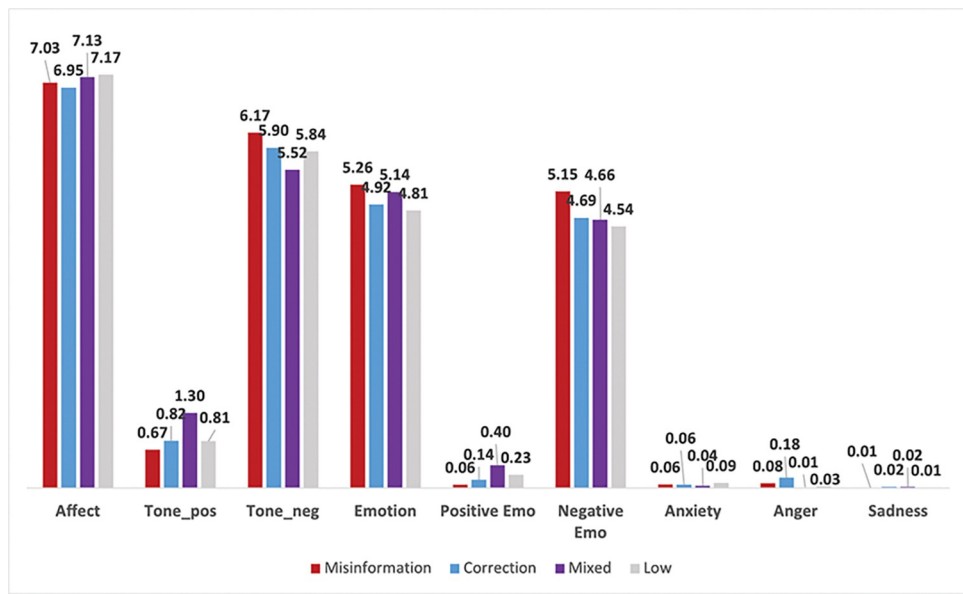

**Fig 4. Average percentage of affect and emotional sentiment across misinformation classifications.**

For sentiment categories measuring social and health-related topics, correction topics on average scored higher than misinformation topics for *Prosocial* (0.29% vs 0.18%, $p < .001$), *Moral* (0.32% vs 0.24%, $p < .001$), *Illness* (4.21% vs 4.18%, $p < .001$), and *Risk* (0.23% vs 0.16%, $p < .001$) sentiments. Misinformation topics scored higher on *Conflict* (.26% vs .24%, $p = .013$), *Health* (4.80% vs 4.62%, $p < .001$), and *Death* (0.33% vs 0.12%, $p < .001$) related words compared to corrections. When compared against the averages of all other misinformation classifications, the mixed topic scored highest in *Moral* (0.48%) and *Risk* (0.52%) sentiments.

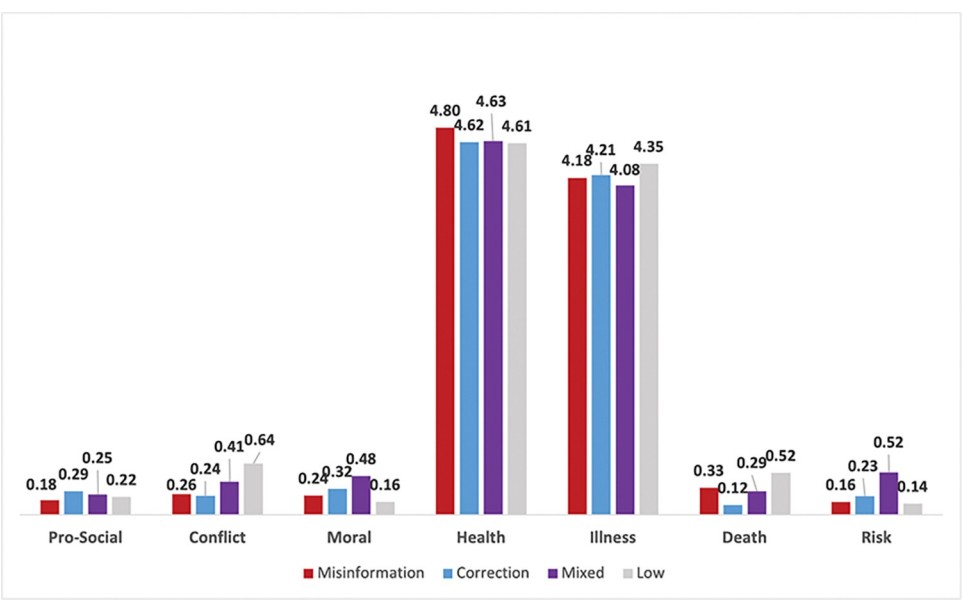

**Fig 5. Average percentage of social and health-related sentiment across misinformation classifications.**

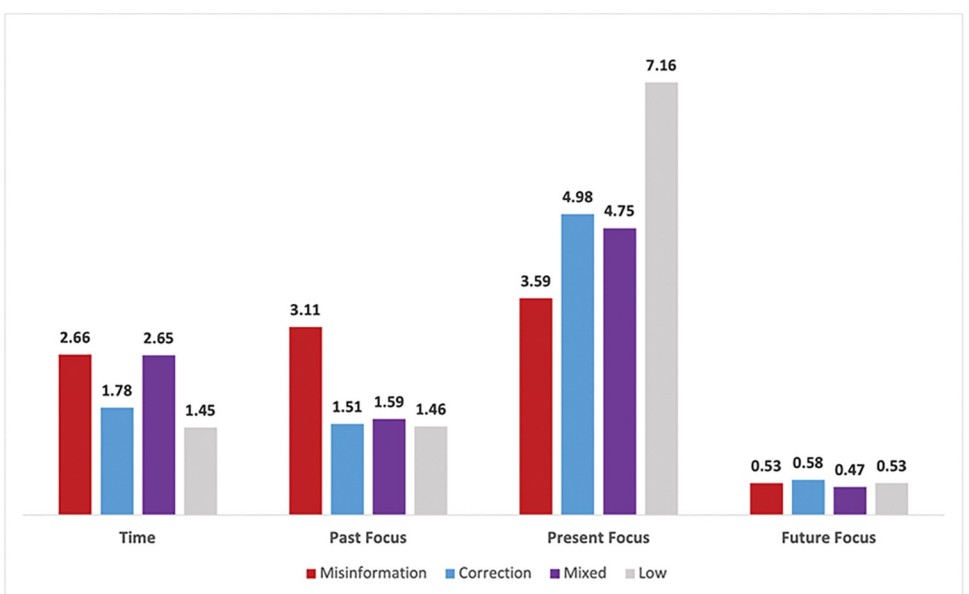

**Fig 6. Average percentage of time-related sentiment across misinformation classifications.**

Lastly, among time-related sentiment categories misinformation topics on average scored higher than all other classifications for general *Time* (2.66%) and *Past-focused* (3.11%), but scored lowest for *Present-focused* (3.59%) and lower than corrections for *Future-focused* (0.53% vs 0.58%, $p < .001$) sentiments. Topics low in both misinformation and corrections scored the highest (7.16%) for *Present-focused* words compared against all other classifications ($p < .001$).

### 3.3 Analysis of user account affiliations

Accounts that composed the top ten most retweeted tweets associated with each topic cluster were grouped together by misinformation classification type, as seen in **Figs 7** and **8**. Accounts were classified as a spreader of misinformation or corrections based on whether they posted tweets from the respective categories as labeled from deductive coding. See **S1 Table** for account information broken out by each topic cluster. Accounts that were engaged in misinformation discourse were the most likely to be suspended (36.6%) or deleted (18.3%) after two years (the time of analysis) from posting their tweets compared to correction discourse where only 2.8% of accounts were suspended. Since a suspended account refers to when the platform kicks off a user's account, these results reflect actions taken by Twitter towards addressing misinformation discourse.

Of the occupational affiliations for most retweeted users, there were 17 medical affiliates / scientists, 27 employed by the media, 7 government officials, and 2 religious leaders. Among accounts engaged in misinformation discourse, there were some occupational affiliations detected with the media (5.6%), medical affiliates / scientists (4.2%), and religious leaders (1.4%). There was no involvement from government officials detected in the misinformation discourse. However, there were relatively more professional affiliations identified in correction discourse where 33.3% of accounts were affiliated with the media, 22.2% were medical affiliates, and 8.3% were government officials. No religious leaders were associated with correction discourse. These findings show that while there were some media and medical affiliates spreading misinformation, the vast majority focused on correcting false information.

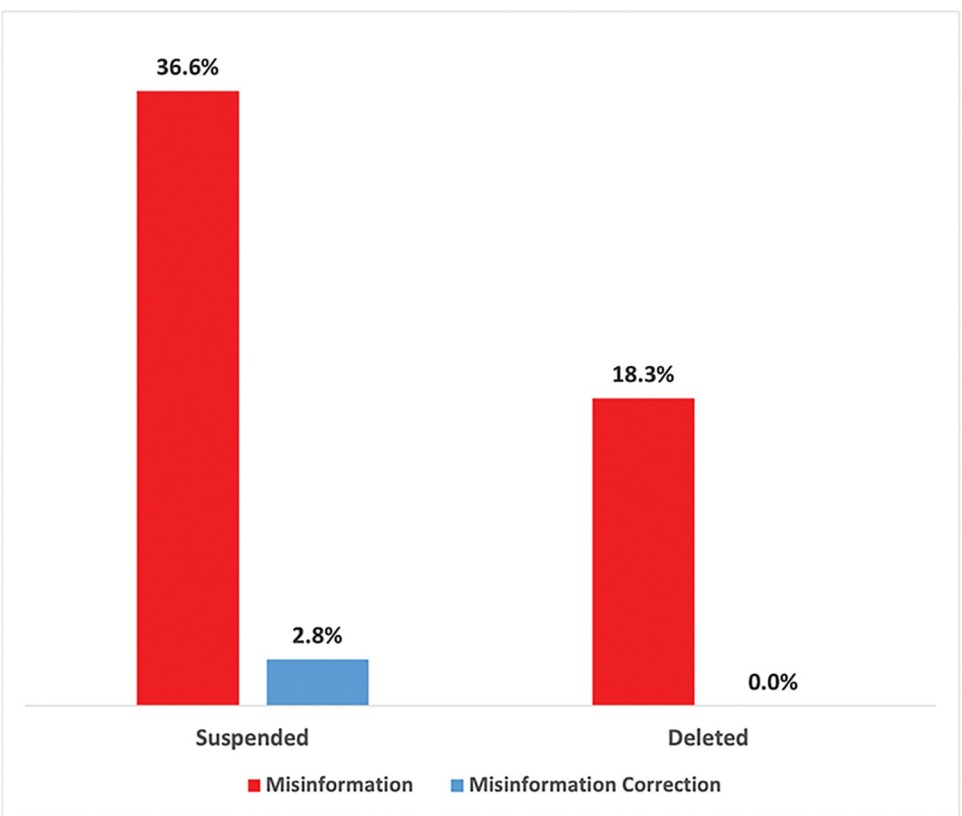

**Fig 7. Most retweeted accounts: Suspended of deleted by misinformation classification.**

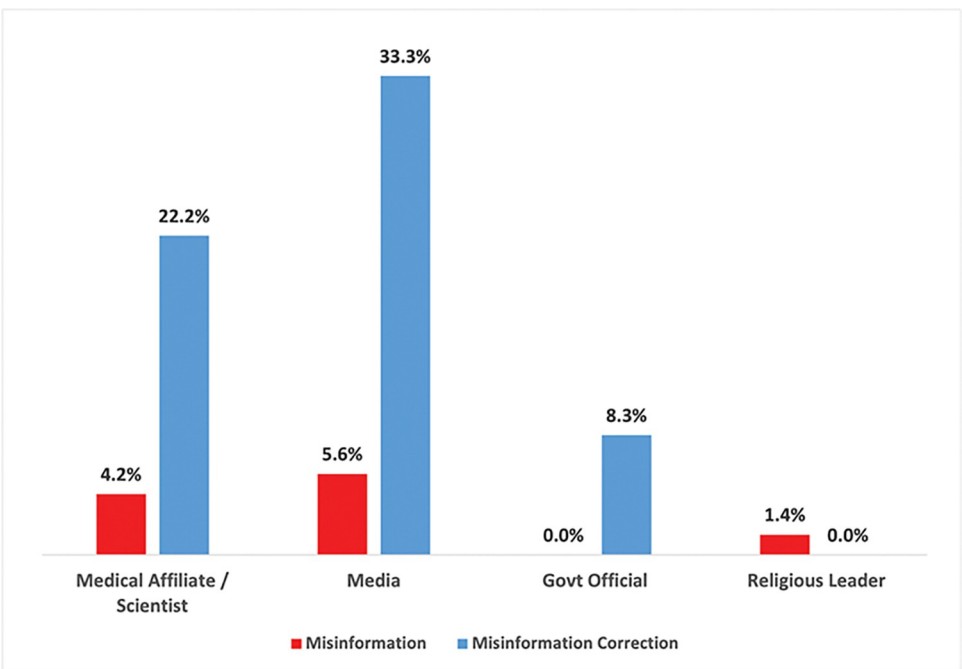

**Fig 8. Most retweeted accounts: Affiliations by misinformation classification.**

## 4. Discussion

This study collected tweets related to 5G conspiracy theories and applied both NLP and qualitative content coding to characterize a large-scale online discourse. Sentiment analysis results show that misinformation and conspiracy discourse was more likely to use language that is analytic and references social status, death, conflict, and health. This discourse also scored higher in general emotion, negative emotion, and past-orientation. In contrast, discourse that challenged false information was more likely to use language related to cognitive processes, positive emotions, anger, prosocial tendencies, morality, illness, risk, and be future-oriented. These results reveal that correction topics, when compared to misinformation and conspiracy discourse, are more likely to use explanatory words related to causative arguments (e.g., "because", "how") and are more concerned with social relations and health-related consequences. The differences in temporal focus between discourses suggest that misinformation narratives evoke past events more often than corrections, whereas the future-oriented language of correction discourse could reflect concerns regarding health-related consequences to 5G-COVID conspiracy theories. These results also reveal that conspiracy-related discourse is more likely to use combative and negative emotional language, and mention extreme consequences more often than corrections (e.g., using death-related words vs illness-related words). The higher degree of negative and emotional language is consistent with previous studies that used sentiment analysis to examine 5G-COVID conspiracies [43, 80], and provides additional insight into rhetorical strategies used in conspiratorial discourse. Further, these findings are consistent with work examining conspiracy spreaders on Twitter more generally, which found significant differences in negative emotion and death-related words compared to those who engage with scientific content [42].

Results from the inductive coding themes were also detected in related work. More specifically, the themes *Group of Elites* [29, 30, 81], *Anti-vax* [30, 81], *Celebrity* [29], *Religious Figures* [29], *Geography Comparisons* [27], *Videos* [82], *Technology correlations with disease outbreaks* [28], and *Negative health effects* [29] were all identified in other studies across Facebook, Twitter, and Instagram. The consistency in findings shows the utility of the approach used in the current study, and reveals a pattern of uniformity in conspiracist narratives across social media platforms.

### 4.1 Using topic modeling to identify discourse for further investigation

This hybrid approach uses topic modeling to identify influential posts within the corpus across multiple topic areas, making it possible to only need to code a small subset of the total volume of tweets in order to detect prominent themes and narratives within both conspiracy and correction discourses. Additionally, segmenting the discourse into topic clusters and then running sentiment analysis makes it possible to detect smaller subsets of tweets within the larger twitter conversation that are high in emotional affect. This can be useful for quickly identifying emotionally charged conversations that could be prioritized for content assessment from public officials or platforms. For instance, since discourse within misinformation topics 0, 6, and 13 are highest in negative emotion, anger and conflict sentiments, as seen in **Table 4**, these might be of relevance to those interested in tracking or preventing mobilization responses that lead to consequences in the physical world or lead to "offline harm" (such as deciding to burn down a cell tower). When assessing the inductive coding themes related to these topics, results show that themes concerning viral videos, negative health effects, and the Chinese technology company Huawei are associated with these sentiments. This could lead to follow-up analyses examining the users who lead these conversations, and can help prioritize which narratives within a conspiracy discourse might be most harmful to the public, especially in cases such as Huawei that could be linked to xenophobic attitudes towards Chinese and Asian Americans.

Another topic of interest is cluster 8, which is associated with the anti-vax theme and high in health and illness sentiments. Despite the timeframe of the study being late March to early April, where for those in the US and most English-speaking countries COVID-19 was still an emerging pandemic, it is surprising to encounter anti-vax sentiment towards the COVID-19 vaccine almost nine months before the announcement of the first approved vaccine. Topic 16, which has inductive coding themes related to causative explanations and medical authorities, and high in sentiment scores for *Anxiousness* and *Pro-social* language, could also be of interest for those looking to identify persuasive strategies used by online conspiracy spreaders. In this discourse, the combination of credibility signifiers reflected in scientific terminology and pro-social language could make messages more persuasive to those seeking certainty and safety during an unknown health crisis or searching for more credible sources of information to fill an existing information gap.

It is also possible to track reactions to misinformation discourse by examining themes prominent in correction topics, where strategies using causative explanations and calls to medical authorities appear to be a common strategy used in corrections. However, the discourse from the corrections did not directly address many of the narratives prevalent in misinformation conversations (e.g., *Huawei*, *Anti-vax*) with the most specific talking-points focusing on criticizing celebrities or religious leaders for spreading conspiracies as also detected by Honcharov et al. in a separate study examining anti-vaccination hashtags of public figures on Twitter [82]. Not addressing specific conspiracist narratives could tamper the effectiveness of correction strategies by not clarifying the information gaps that are capitalized on by misinformation spreaders, especially during times of uncertainty. This is particularly important in light of the findings from the affiliations analysis of the most retweeted users, which showed that media figures, medical scientists, and government officials were more likely to be involved in corrections discourse. Since these accounts are more likely to have higher perceived credibility and a greater reach in audience compared to the average user, it is important that emerging conspiracist narratives are identified early in order to design more effective counter-messaging strategies.

## 4.2 Drawbacks of solely relying on machine learning approaches

Machine learning approaches for detecting misinformation (or classifying categories within online discourse more generally) have continued to evolve over the years. Recent efforts to improve accuracy of misinformation classifiers have found success using theoretical frameworks based on human information processing to guide feature selection [83], combining features from multiple modalities (i.e., text and visual) [84], and including features related to user response to post and message sources [85]. Multiple algorithms have also been evaluated, ranging from traditional logistic regression models and ensemble approaches such as voting and bagging classifiers [86] to advanced models using deep convolutional neural networks and knowledge graphs that use word embeddings instead of requiring feature selection [87, 88].

In response to the high accuracy of machine learning classifiers at detecting misinformation, there have been calls for action in recent years for researchers to make publicly available textual data with labeled fake news articles to build comprehensive training datasets for misinformation classification [89–91]. These findings, in addition to the prominence of advanced LLMs such as chatGPT [92, 93], make it tempting to infer that AI technologies can fully address the issue of large-scale content coding without the use of humans. However, as previously discussed, an important drawback of machine learning approaches is that they are generally limited to detecting fake news within specific contexts. Since misinformation topics and conspiracy theories are always evolving, NLP approaches are unable to generalize to novel

situations that are not already documented. For example, classifiers developed in response to the misinformation proliferation surrounding the 2016 US presidential election would not be able to adequately detect COVID-related misinformation since the relevant keywords do not correspond across topics. Maintaining up-to-date classifier models would also be difficult to achieve during the COVID-19 pandemic, where scientific consensus and understanding of the virus have shifted multiple times from its early stages, and continues to evolve even over three years after its initial outbreak. It is also difficult for machine learning approaches to detect instances where outdated information or findings that were considered accurate at the time of publication are then used misleadingly in contemporary scenarios, such as in the case surrounding misinformation promoting the use of hydroxychloroquine for treating COVID-19 [56, 58].

While building effective classifier models for 5G COVID conspiracies is possible during 2023 after being well documented, only human coders had the capability to accurately recognize text that contains these narratives when they first emerged online in 2020. We intended to address these limitations by demonstrating the general utility of a hybrid approach that incorporates human coders to take advantage of the efficiencies machine learning techniques provide researchers when working with big datasets while still accounting for contextual nuance from using qualitative approaches. Even though the definition of misinformation may vary in other discussions (e.g., politics, climate change, other social issues), the general principles of the methodology described could be leveraged to provide more up-to-date and richer contextual insights into how these conspiracy-related discourses evolve over time.

## 4.3 Future directions in infodemiology

In order to properly make sense of findings generated from large scale communication networks, it is important that we also advance our conceptual understandings of information transmission dynamics. Other fields such as cognitive science and human-computer interactions (HCI) have developed frameworks for scientifically examining information as a phenomenon, which can be incorporated into future infodemiology research. One such theory is information foraging, which is based on ecological models of food scavenging behaviors where online users are considered "foragers" who balance the value gained from finding new information with the time cost needed to obtain it [94–98]. The types of analyses conducted using an information foraging framework include the following: information patch models that assess engagement activities in environments where information is encountered in clusters (e.g., webpages), information scent models that assess how information value is evaluated from proximal cues (e.g., titles and images on a website), and information diet models that examine decisions concerning the selection and pursuit of information items [94]. Future infodemiological work can design behavioral experiments using information scent models to test how cues on sites or social media posts influence safety perceptions of potentially dangerous transactions such as illicit drug purchasing, and adapt information diet models to examine how users evaluate the truthfulness of health-claims across online environments.

Another relevant theoretical framework is distributed cognition, which extends human intelligence beyond the boundaries of individual actors to encompass interactions between people and resources within the environment [99–101]. According to this framework, social organization determines the way information flows through groups [100], and more recent work emphasizes the ways cognition is distributed over a vast array of social, institutional, political, and technological systems that are shaped by, and shape, the individuals who develop and operate within them [99, 102]. Within the context of social media environments, where topics of discourse are constantly shifting and information is transmitted from a wide array of

sources including other users, celebrities, institutions, and political figures, this framework can make sense of how narratives evolve by examining the interplay between the broadcaster of a message and their audiences' reaction, engagement, and retransmission of that message. The theory of distributed cognition can also guide research questions that distinguish between message transmission due to an individual's personal beliefs and propagation driven by conformity to other users. Overall, both information foraging and distributed cognition frameworks can be adapted in future work to generate deeper insights concerning health-related information seeking behaviors, enrich interpretations from social network analysis, and guide the design of interventions targeting misinformation propagation.

### 4.4 Limitations

Content coding took place several months after the initial timeframe of the study. While having the time gap allowed us to assess which accounts were deleted or suspended since the initial 5G discourse, we were unable to code for affiliations of those deleted users using publicly available profile data. As stated in the methods section, the top 10 retweeted tweets do not account for every tweet associated within a topic cluster. While the most retweeted tweets account for a substantive proportion of the topic cluster's tweet volume, and in most cases a majority, there is still some level of uncertainty when characterizing the discourse even if the uncoded tweets share textual similarity to the coded posts. Additional measures such as the use of sentiment analysis, which in this study was applied to the full topic clusters, can also mitigate these concerns since they account for information provided in the uncoded tweets. Finally, though false information can be categorized as "misinformation," "disinformation," "mal-information," and "conspiracy" based on intent and content, this study did not differentiate between these categories, opting to call all false information, regardless of intent, "misinformation." This lack of differentiation limits the study's ability to identify potential differences in rhetoric associated with the nuances of false information dissemination.

## 5. Conclusion

The advancement of communication technologies and the continued emergence of new social media platforms present difficult challenges for researchers looking to investigate the highly dynamic information ecosystem of the 21st century. Fortunately, these same rapid advancements in technology can also be harnessed by researchers as powerful tools to navigate these complex environments. However, as demonstrated in the current paper, the human perspective is equally crucial in this line of work to compensate for the shortcomings that artificial intelligences have toward understanding human endeavors. Due to the rapid pace of modern discourse, words that can be key identifiers for a dangerous conspiracy in one context can be completely irrelevant in a different grouping of text. Within these online conversations, where the boundary between signal and noise is constantly shifting due to emerging and continually evolving narratives, it is crucial to recruit the signal detection capabilities of both machine learning models and human beings to adequately address current and future misinformation challenges now endemic in our global information society.

## Supporting information

**S1 Checklist. STROBE statement—Checklist of items that should be included in reports of observational studies.**
(DOCX)

**S1 Appendix. Hybrid approach for characterizing social media discourse using machine learning and qualitative methodologies.**
(PDF)

**S1 Table. Most retweeted accounts associated with each BTM cluster.**
(PDF)

## Acknowledgments

The authors would like to thank our collaborators in the Cognitive Media Lab at UC San Diego (cognitivemedialab.ucsd.edu) for the productive discussions about online conspiracy discourse that helped inform the current manuscript.

## Author Contributions

**Conceptualization:** Michael Robert Haupt, Michelle Chiu, Joseline Chang.

**Data curation:** Zoe Li.

**Formal analysis:** Michael Robert Haupt, Zoe Li.

**Methodology:** Michael Robert Haupt.

**Project administration:** Joseline Chang.

**Supervision:** Raphael Cuomo, Tim K. Mackey.

**Visualization:** Michael Robert Haupt.

**Writing – original draft:** Michael Robert Haupt.

**Writing – review & editing:** Michael Robert Haupt, Michelle Chiu, Raphael Cuomo, Tim K. Mackey.

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
