## [Decision Letter · Decision Letter 0]

31 Jul 2023

PONE-D-23-14668Detecting Nuance in Conspiracy Discourse: Advancing Methods in Infodemiology and Communication Science with Machine Learning and Qualitative Content CodingPLOS ONE

Dear Dr. Haupt,

Thank you for submitting your manuscript to PLOS ONE. After careful consideration, we feel that it has merit but does not fully meet PLOS ONE’s publication criteria as it currently stands. Therefore, we invite you to submit a revised version of the manuscript that addresses the points raised during the review process. The reviewers have offered valuable insights and constructive criticism aimed at strengthening your manuscript. Their expertise and perspectives will undoubtedly contribute to refining your research and ensuring its significance within the field. It is crucial that you carefully consider their comments and respond to each point thoughtfully.

Information regarding ethical challenges in digitalization could be found in:

https://www.mdpi.com/2073-431X/12/6/114.

Please review and evaluate the requested work to determine whether they are relevant and should be cited. It is not a requirement to cite these works. We appreciate your attention to this request.

We look forward to receiving your revised manuscript.

Regards,

Jasna Karacic Zanetti, Ph.D

Academic Editor 

Plos One

2. In your Methods section, please include additional information about your dataset and ensure that you have included a statement specifying whether the collection and analysis method complied with the terms and conditions for the source of the data.

“Authors TKM and ZL are employees of the startup company S-3 Research LLC. S-3 Research is a startup funded and currently supported by the National Institutes of Health – National Institute of Drug Abuse through a Small Business Innovation and Research contract for opioid-related social media research and technology commercialization. TKM is also the CEO and a member of S-3 Research LLC with ownership.  Author reports no other conflict of interest associated with this manuscript.”

Reviewers' comments:

Reviewer's Responses to Questions

**Comments to the Author**

1. Is the manuscript technically sound, and do the data support the conclusions?

Reviewer #1: Yes

Reviewer #2: Yes

Reviewer #3: Yes

Reviewer #4: Partly

2. Has the statistical analysis been performed appropriately and rigorously? 

Reviewer #1: Yes

Reviewer #2: Yes

Reviewer #3: Yes

Reviewer #4: Yes

3. Have the authors made all data underlying the findings in their manuscript fully available?

Reviewer #1: Yes

Reviewer #2: Yes

Reviewer #3: Yes

Reviewer #4: Yes

4. Is the manuscript presented in an intelligible fashion and written in standard English?

Reviewer #1: Yes

Reviewer #2: Yes

Reviewer #3: Yes

Reviewer #4: Yes

5. Review Comments to the Author

Reviewer #1: The manuscript ensures that the message conveyed is clear, concise, and easily comprehensible to the intended audience. Its well-written manuscript uses appropriate grammar, spelling, punctuation, and vocabulary to convey the author's ideas effectively. It also follows a logical structure that enables readers to follow the flow of ideas easily.

Reviewer #2: 1. Summary of research and overall impression

The manuscript is a well written study of misinformation recognized in online platforms such as Twitter. It takes a thorough view of various approaches to misinformation detection and classification and also examines the societal need for this activity, a field termed as Infodemiology in the health context.

2. Major Issues

The work has some excessively wide-ranging phrases and general statements, for example, ``Since science itself is an iterative process..." which result in a slightly long manuscript and could be pruned. Moreover, such statements are not strictly necessary, apparently.

Minor Issues

Details such as the specific script/algorithm used for analysis in each Table could be presented in Supplementary Data or in an Appendix to make the work more replicable by the readership.

As an additional point, as the manuscript is long, adding a `table of contents' type element right at the start might be useful for the reader to get a bird's-eye view of all the sections and subsections; but journal format guidelines have the last word on this.

3. Overall Feedback

The work is well referenced and the presentation conveys the impression of thoroughness.

Reviewer #3: This is an excellent study and manuscript describing the use of a hybrid method for detecting conspiracy tweets linking 5G technology to COVID-19. I especially appreciated the authors’ suggested applications of their findings in the Discussion. A few points are provided below for the authors to consider.

• The Introduction provides an excellent and erudite background on infodemiology and content coding. It is longer than is standard, but perhaps worth keeping given the broad audience of PLOS One.

• Given the large amount of information scientists encounter daily, many will only see the abstract of this study. It is therefore important to include some findings or results in the abstract. I urge the authors to add at least two sentences describing the outcome of their analysis of tweets linking 5G to COVID-19; several of the background sentences can be omitted, if needed, to accommodate the word limit. Even though the authors position the study as a showcase for their methods, it seems important to demonstrate the novel insights that the method provides. Similarly, the authors state that “this study showcases . . . .” in the abstract, without describing the study. These changes will help rectify the current feel of over-promoting the method instead of letting the data/outcomes speak for itself/themselves.

• COVID-19 or coronavirus should be listed as a keyword, as it does not appear in the abstract or title. Consider added 5G as a keyword also, although that does appear in the abstract.

• A description of the units shown should be added to each graph or in the figure captions. This could be accomplished by labeling the Y axis of each graph.

• This study—including both the methods and findings—are of broad societal interest and may attract a wide readership. For this reason, I suggest that the authors add a simplified (flow?) diagram of the general methods used, highlighting the novel features and/or outcomes it permits.

• I suggest deleting the last sentence of the Conclusion, as it seems a bit “preachy” for a scientific paper and may work better in an editorial.

Reviewer #4: This study investigated conspiracy discourse related to 5G wireless technology by using a combination of NLP (i.e., topic modeling and sentiment analysis) and qualitative content analysis.

1. The abstract should reflect the main findings of the current study.

2. The first paragraph concludes with a call to advance methodologies that address the ongoing proliferation of misinformation that is overwhelming the social media ecosystem. I got a little lost here. This conclusion seems disconnected from the main reasoning of this paragraph. Why can't the aforementioned infodemiology applications address the current problems? What areas in particular need to be advanced?

3. The authors should clearly distinguish between misinformation and conspiracies. They are two distinct concepts, but the authors used them interchangeably throughout the manuscript.

4. I have some concerns over the generalizability of the current study. The NLP of this study generally evolves around the 5G-COVID conspiracy. How could the findings be generalized to misinformation detection in other disciplines, say politics, climate change, etc.?

5. Please elaborate on the relativistic approach: define it and justify its use in the current context.

6. The first paragraph on page 9 appears disjointed as it initially discusses three modes of data interpretation, then transitions to discussing inductive and deductive analysis. It would be beneficial to establish a coherent connection between these two sets of approaches to ensure the flow of ideas and the logical progression of the argument.

7. I guess most of the audience of this paper should get some foundational knowledge, like what inductive and deductive coding are, as well as their pros and cons. So the authors could cut this part.

8. Deductive coding is a coding process intended to test whether data coincides with existing assumptions, theories, or hypotheses. I agree with that. But I don’t see any assumptions or hypotheses that the authors sought to test.

9. My understanding is that this study summarized the topic and linguistic characteristics of 5G-related conspiracies on Twitter using topic modeling, LIWC, and manual content analysis. First, it is not about detection. The paper did not present any algorithms or computations that could be used to detect falsehoods in the future. Second, these methods are not quite new (see below)

Rashkin, H., Choi, E., Jang, J. Y., Volkova, S., & Choi, Y. (2017, September). Truth of varying shades: Analyzing language in fake news and political fact-checking. In Proceedings of the 2017 conference on empirical methods in natural language processing (pp. 2931-2937).

Zhou, C., Li, K., & Lu, Y. (2021). Linguistic characteristics and the dissemination of misinformation in social media: The moderating effect of information richness. Information Processing & Management, 58(6), 102679.

Massey, P. M., Kearney, M. D., Hauer, M. K., Selvan, P., Koku, E., & Leader, A. E. (2020). Dimensions of misinformation about the HPV vaccine on Instagram: Content and network analysis of social media characteristics. Journal of medical Internet research, 22(12), e21451.

6. PLOS authors have the option to publish the peer review history of their article (what does this mean?). If published, this will include your full peer review and any attached files.

Reviewer #1: No

Reviewer #2: No

Reviewer #3: No

Reviewer #4: No

---

## [Author Response · Author response to Decision Letter 0]

7 Sep 2023

To: Dr. Jasna Karacic Zanetti, Academic Editor, PLOS ONE

Re: Revised Manuscript, “Detecting Nuance in Conspiracy Discourse: Advancing Methods in Infodemiology and Communication Science with Machine Learning and Qualitative Content Coding”

Date: September 7th, 2023

Dear Prof. Zanetti, 

Thank you for your July 31st, 2023 editorial and reviewer comments for the manuscript titled “Detecting Nuance in Conspiracy Discourse: Advancing Methods in Infodemiology and Communication Science with Machine Learning and Qualitative Content Coding” (Manuscript ID PONE-D-23-14668).

We thank you for your continued interest in our submission along with helpful reviewer comments. As directed by your correspondence, we are submitting a revised manuscript with changes highlighted in the text and this cover letter that details how we have addressed all reviewer comments and suggestions. Our responses are highlighted in yellow, with quoted text from the revised manuscript in 11-point font. The convention of pX, ¶A (i.e., page X, paragraph A) is adopted to indicate from where in the revised manuscript the text we reproduce arises.

Below, please find the editor and reviewer’s comments (in bold) and how each is addressed in a point-by-point response in the revised manuscript. Changes to the manuscript have also been highlighted in yellow in the revision submitted.

We thank you for the opportunity to submit this revised manuscript and look forward to your comments.

Journal Requirements

Author Response: Thank you for your feedback about the formatting of our manuscript. We have updated our manuscript so that it is now aligned with PLOS ONE’s style requirements.

2. In your Methods section, please include additional information about your dataset and ensure that you have included a statement specifying whether the collection and analysis method complied with the terms and conditions for the source of the data.

Author Response: Thank you for your comment for more information concerning our data collection and analysis method. We have added language to the Methods section detailing that our collection was compliant with the terms and conditions associated with publicly available data and terms per the Twitter API at the time of the study.

2.1 Data Collection and Analysis Overview

A total of 256,562 tweets were collected from the public streaming Twitter API per the terms available at the time of the study using keywords “5G” and covid-related words such as “coronavirus” and “covid-19” between March 25th to April 3rd 2020. We chose this time frame as it represents a period when the 5G conspiracy theory first became prominent, as shown in the spike in volume for “5G” posts in Figure 1. All personal identifiable information from tweets was removed in the reporting of the results to preserve anonymity. We note that due to the change in ownership, API policies, and name of the platform (Twitter has been renamed “X”), the terms and conditions of the streaming API used for data collection for this study are now longer applicable for current studies. IRB approval was not required as all data collected in this study was available in the public domain and results from the study have been deidentified and anonymized. The dataset and R syntax used to generate the results can be found at the following Open Science Framework (OSF) link: https://bit.ly/5G_Conspiracies. p13, ¶3 

“Authors TKM and ZL are employees of the startup company S-3 Research LLC. S-3 Research is a startup funded and currently supported by the National Institutes of Health – National Institute of Drug Abuse through a Small Business Innovation and Research contract for opioid-related social media research and technology commercialization. TKM is also the CEO and a member of S-3 Research LLC with ownership. Author reports no other conflict of interest associated with this manuscript.”

Author Response: We have included the previously mentioned phrase into our disclosure statement to clarify that author affiliation with S-3 Research does not alter our adherence to PLOS ONE policies on sharing data and materials. 

Disclosure Statement

TKM and ZL are employees of the startup company S-3 Research LLC. S-3 Research is a startup funded and currently supported by the National Institutes of Health – National Institute of Drug Abuse through a Small Business Innovation and Research contract for opioid-related social media research and technology commercialization. TKM is also the CEO and a member of S-3 Research LLC with ownership. This does not alter our adherence to PLOS ONE policies on sharing data and materials. Author reports no other conflict of interest associated with this manuscript. 

 

REVIEWER #1

The manuscript ensures that the message conveyed is clear, concise, and easily comprehensible to the intended audience. Its well-written manuscript uses appropriate grammar, spelling, punctuation, and vocabulary to convey the author's ideas effectively. It also follows a logical structure that enables readers to follow the flow of ideas easily.

Author Response: We thank the reviewer for their positive comments regarding our manuscript. 

REVIEWER #2

The manuscript is a well written study of misinformation recognized in online platforms such as Twitter. It takes a thorough view of various approaches to misinformation detection and classification and also examines the societal need for this activity, a field termed as Infodemiology in the health context.

2. Major Issues

The work has some excessively wide-ranging phrases and general statements, for example, ``Since science itself is an iterative process..." which result in a slightly long manuscript and could be pruned. Moreover, such statements are not strictly necessary, apparently.

Author Response: Thank you for your feedback concerning the phrasing used in our manuscript and overall length. In response we have restructured our introduction and discussion in order to remove repetitive language and overly general statements. We also rephrased sentences throughout the manuscript to make the language more concise and clearer when possible. Changes are numerous, please see highlighted changes in the revised manuscript submitted.

Minor Issues

Details such as the specific script/algorithm used for analysis in each Table could be presented in Supplementary Data or in an Appendix to make the work more replicable by the readership.

Author Response: Thank you for your feedback about making our analysis scripts accessible for replicability. We have added the R-script used for analysis into our open science framework link, as well as referencing the availability of the syntax in the revised methods section. See revised language below:

2.1 Data Collection and Analysis Overview

A total of 256,562 tweets were collected from the public streaming Twitter API per the terms available at the time of the study using keywords “5G” and covid-related words such as “coronavirus” and “covid-19” between March 25th to April 3rd 2020. We chose this time frame as it represents a period when the 5G conspiracy theory first became prominent, as shown in the spike in volume for “5G” posts in Figure 1. All personal identifiable information from tweets was removed in the reporting of the results to preserve anonymity. We note that due to the change in ownership, API policies, and name of the platform (Twitter has been renamed “X”), the terms and conditions of the streaming API used for data collection for this study are now longer applicable for current studies. IRB approval was not required as all data collected in this study was available in the public domain and results from the study have been deidentified and anonymized. The dataset and R syntax used to generate the results can be found at the following Open Science Framework (OSF) link: https://bit.ly/5G_Conspiracies. p13, ¶3

As an additional point, as the manuscript is long, adding a `table of contents' type element right at the start might be useful for the reader to get a bird's-eye view of all the sections and subsections; but journal format guidelines have the last word on this.

Author Response: Thank you for your comment about adding a table of contents element to our manuscript. We will reach out to the editor in order to see the feasibility of adding this feature. 

REVIEWER #3

This is an excellent study and manuscript describing the use of a hybrid method for detecting conspiracy tweets linking 5G technology to COVID-19. I especially appreciated the authors’ suggested applications of their findings in the Discussion. A few points are provided below for the authors to consider.

• The Introduction provides an excellent and erudite background on infodemiology and content coding. It is longer than is standard, but perhaps worth keeping given the broad audience of PLOS One.

Author Response: Thank you for your feedback concerning the introduction section. In response to reviewer’s suggestions and those of other reviewers, we have edited the Introduction to maintain the main points in the previous version while trimming excess statements and rephrasing the language in order to make it clearer and more concise when possible. Changes are numerous, please see highlighted changes in the revised manuscript submitted.

• Given the large amount of information scientists encounter daily, many will only see the abstract of this study. It is therefore important to include some findings or results in the abstract. I urge the authors to add at least two sentences describing the outcome of their analysis of tweets linking 5G to COVID-19; several of the background sentences can be omitted, if needed, to accommodate the word limit. Even though the authors position the study as a showcase for their methods, it seems important to demonstrate the novel insights that the method provides. Similarly, the authors state that “this study showcases . . . .” in the abstract, without describing the study. These changes will help rectify the current feel of over-promoting the method instead of letting the data/outcomes speak for itself/themselves.

Author Response: Thank you for your important comment about featuring our results within the abstract. In response we have included in the abstract the major results from the study, including sentiment analysis and inductive coding in order to make the insights generated from our approach more explicit. We have also edited the abstract to remove any language that might be interpreted as self-promoting, such as the language suggested by reviewer. See revised abstract below:

Abstract

The spread of misinformation and conspiracies has been an ongoing issue since the early stages of the internet era, resulting in the emergence of the field of infodemiology (i.e., information epidemiology) which investigates the transmission of health-related information. Due to the high volume of online misinformation in recent years, there is a need to continue advancing methodologies in order to effectively identify narratives and themes. While machine learning models can be used to detect misinformation and conspiracies, these models are limited in their generalizability to other datasets and misinformation phenomenon, and are often unable to detect implicit meanings in text that require contextual knowledge. In order to rapidly detect evolving conspiracist narratives within high volume online discourse while identifying nuanced themes requiring the comprehension of subtext, this study describes a hybrid methodology that combines natural language processing (i.e., topic modeling and sentiment analysis) with qualitative content coding approaches to examine conspiracy discourse related to 5G wireless technology and COVID-19 on Twitter (currently known as ‘X’). Discourse that focused on correcting 5G conspiracies was also analyzed for comparison. Sentiment analysis shows that conspiracy-related discourse was more likely to use language that was analytic, reference social status, combative, expressed negative emotions, and be past-oriented. Corrections discourse was more likely to use words reflecting cognitive processes, prosocial relations, health-related consequences, and future-oriented language. Inductive coding identified conspiracist narratives related to global elites, anti-vax sentiment, medical authorities, religious figures, and false correlations between technology advancements and disease outbreaks. Further, the corrections discourse did not address many of the narratives prevalent in conspiracy conversations. This paper aims to further bridge the gap between computational and qualitative methodologies by demonstrating how both approaches can be used in tandem to emphasize the positive aspects of each methodology while minimizing their respective drawbacks. 

• COVID-19 or coronavirus should be listed as a keyword, as it does not appear in the abstract or title. Consider added 5G as a keyword also, although that does appear in the abstract.

Author Response: Thank you for your suggestion concerning our keywords. In response we have now added the terms ‘COVID-19’ and ‘5G’. 

• A description of the units shown should be added to each graph or in the figure captions. This could be accomplished by labeling the Y axis of each graph.

Author Response: Thank you for your comment about specifying the units used in Y axis for our figures. In response we have added to the figure caption the phrase “Average percentage of…” to specify what the Y axis is depicting. 

• This study—including both the methods and findings—are of broad societal interest and may attract a wide readership. For this reason, I suggest that the authors add a simplified (flow?) diagram of the general methods used, highlighting the novel features and/or outcomes it permits.

Author Response: Thank you for your suggestion about including a flow chart that further illustrates the methods used in our study. We have now included a chart in the appendix and referenced it in the Introduction section. 

• I suggest deleting the last sentence of the Conclusion, as it seems a bit “preachy” for a scientific paper and may work better in an editorial.

Author Response: Thank you for your feedback concerning the language used in our Conclusion. In response we have edited the language so that it focuses back on the advantages of using the approach featured in the manuscript and deleted the last sentence as suggested. See revised language below:

5. Conclusion

The advancement of communication technologies and the continued emergence of new social media platforms present difficult challenges for researchers looking to investigate the highly dynamic information ecosystem of the 21st century. Fortunately, these same rapid advancements in technology can also be harnessed by researchers as powerful tools to navigate these complex environments. However, as demonstrated in the current paper, the human perspective is equally crucial in this line of work to compensate for the shortcomings that artificial intelligences have towards understanding human endeavors. Due to the rapid pace of modern discourse, words that can be key identifiers for a dangerous conspiracy in one context can be completely irrelevant in a different grouping of text. Within these online conversations, where the boundary between signal and noise is constantly shifting due to emerging and continually evolving narratives, it is crucial to recruit the signal detection capabilities of both machine learning models and human beings to adequately address current and future misinformation challenges now endemic in our global information society. p41, ¶2 

REVIEWER #4

This study investigated conspiracy discourse related to 5G wireless technology by using a combination of NLP (i.e., topic modeling and sentiment analysis) and qualitative content analysis.

1. The abstract should reflect the main findings of the current study.

Author Response: Thank you for your important comment about featuring our results within the abstract. In response we have included in the abstract the major results from the study, including sentiment analysis and inductive coding in order to make the insights generated from our approach more explicit. See revised abstract below. 

Abstract

The spread of misinformation and conspiracies has been an ongoing issue since the early stages of the internet era, resulting in the emergence of the field of infodemiology (i.e., information epidemiology) which investigates the transmission of health-related information. Due to the high volume of online misinformation in recent years, there is a need to continue advancing methodologies in order to effectively identify narratives and themes. While machine learning models can be used to detect misinformation and conspiracies, these models are limited in their generalizability to other datasets and misinformation phenomenon, and are often unable to detect implicit meanings in text that require contextual knowledge. In order to rapidly detect evolving conspiracist narratives within high volume online discourse while identifying nuanced themes requiring the comprehension of subtext, this study describes a hybrid methodology that combines natural language processing (i.e., topic modeling and sentiment analysis) with qualitative content coding approaches to examine conspiracy discourse related to 5G wireless technology and COVID-19 on Twitter (currently known as ‘X’). Discourse that focused on correcting 5G conspiracies was also analyzed for comparison. Sentiment analysis shows that conspiracy-related discourse was more likely to use language that was analytic, reference social status, combative, expressed negative emotions, and be past-oriented. Corrections discourse was more likely to use words reflecting cognitive processes, prosocial relations, health-related consequences, and future-oriented language. Inductive coding identified conspiracist narratives related to global elites, anti-vax sentiment, medical authorities, religious figures, and false correlations between technology advancements and disease outbreaks. Further, the corrections discourse did not address many of the narratives prevalent in conspiracy conversations. This paper aims to further bridge the gap between computational and qualitative methodologies by demonstrating how both approaches can be used in tandem to emphasize the positive aspects of each methodology while minimizing their respective drawbacks. 

2. The first paragraph concludes with a call to advance methodologies that address the ongoing proliferation of misinformation that is overwhelming the social media ecosystem. I got a little lost here. This conclusion seems disconnected from the main reasoning of this paragraph. Why can't the aforementioned infodemiology applications address the current problems? What areas in particular need to be advanced?

Author Response: Thank you for your feedback concerning the language used in the introduction. Our intent was to preview further discussion of the potential need to examine new methodologies to address misinformation within the field of infodemiology and not to specifically critique prior work. In response we deleted the last sentence of the first paragraph, edited the first paragraph for greater clarity, and restructured the introduction more generally to better contextualize prior infodemiology and machine learning approaches to the misinformation challenge, while also providing more rationale of why new methodologies also warrant further study. Changes are numerous, please see our revised Introduction section in the revised manuscript. 

3. The authors should clearly distinguish between misinformation and conspiracies. They are two distinct concepts, but the authors used them interchangeably throughout the manuscript.

Author Response: Thank you for your comment about distinguishing between misinformation and conspiracies. We have edited the manuscript in the introduction and discussion to include the phrase “misinformation and conspiracies” when referring to discourse containing false information. For our analysis we have continued to just refer to this discourse as misinformation, however, we included language in the methods to explicitly state that we’re using the term misinformation to refer to both concepts for brevity and have also included additional language in the revised Limitations section clearly stating the limitations of this coding approach. See additional explanation below:

2.3 Deductive and Inductive Coding Schemes

In order to characterize highly prevalent misinformation and conspiratorial narratives in the corpus, the top 10 most retweeted tweets from all 20 BTM topic outputs were extracted and manually coded for relevance first using a deductive coding scheme adapted from existing COVID-19 misinformation themes from the literature (Haupt et al., 2021a; Islam et al., 2020), and then coded again using an inductive approach identifying context specific themes related to 5G. While misinformation and conspiracies are distinct concepts, the current study will refer to both as ‘misinformation’ within the analysis for brevity. p17, ¶1

…

4.4. Limitations

Content coding took place several months after the initial timeframe of the study. While having the time gap allowed us to assess which accounts were deleted or suspended since the initial 5G discourse, we were unable to code for affiliations of those deleted users using publicly available profile data. As stated in the methods section, the top 10 retweeted tweets do not account for every tweet associated within a topic cluster. While the most retweeted tweets account for a substantive proportion of the topic cluster’s tweet volume, and in most cases a majority, there is still some level of uncertainty when characterizing the discourse even if the uncoded tweets share textual similarity to the coded posts. Additional measures such as the use of sentiment analysis, which in this study was applied to the full topic clusters, can also mitigate these concerns since they account for information provided in the uncoded tweets. Finally, though false information can be categorized as “misinformation,” “disinformation,” “mal-information”, and some information can also be classified as “conspiracy” based on intent and content, this study did not differentiate between these categories, opting to call all false information, regardless of intent, “misinformation.” This lack of differentiation limits internal reliability and the study’s ability to identify potential differences in rhetoric associated with the nuances of false information dissemination. p40, ¶2

4. I have some concerns over the generalizability of the current study. The NLP of this study generally evolves around the 5G-COVID conspiracy. How could the findings be generalized to misinformation detection in other disciplines, say politics, climate change, etc.?

Author Response: Thank you for your feedback concerning the generalizability of the NLP methods showed in the current study. In response we have added additional language discussing how our approach can be adapted for other topics. We have also included a flow chart diagram in the appendix (Figure A1) to further illustrate how our approach can be adapted for analyzing other discourse topics besides 5G conspiracies. We have also clarified that the Discussion section of the study that results are not necessarily generalizable to misinformation discourse on other pressing social and health topics, but that the general methodological approach could be leveraged in a similar fashion to provide more contextual classification to misinformation specific to these topics. See revised language below: 

Interpreting sentiment scores can be further complicated when accounting for the fact that the discussion topic can also influence the average emotional tone of a discourse. For example, discussions about a pandemic may have higher percentages of negative affect words (e.g., “death”, “tragedy”) compared to lighter conversation topics such as gardening. For this reason it is difficult to determine what is an appropriate threshold for meaningful sentiment scores across topics. While this can limit what researchers can infer from these analyses, this study addresses this limitation by using a relativistic interpretation of sentiment scores. More specifically, discourse will be marked as high in a sentiment category based on whether it is in the 90th percentile of scores within the corpus. Using the 90th percentile as a threshold marker allows us to account for the specific context of the 5G discourse when making judgements for determining what is a high level of sentiment. This approach is also adaptable across a wide array of topics since percentiles indicate which cases are high and low for a given metric based on the sample distribution. To further illustrate this point, a post containing 5% of death-related words may be in the 95th percentile for discourse about gardening, making it a “high” amount, but be within the 50th percentile for pandemic-related discourse, making it a typical percentage within the context of that corpus. Additionally, this study incorporates a qualitative coding approach to account for contextual information that is typically overlooked in sentiment analysis but can be recognized by a human coder. p8, ¶2

…

While building effective classifier models for 5G COVID conspiracies is possible during 2023 after being well documented, only human coders had the capability to accurately recognize text that contains these narratives when they first emerged online in 2020. We intended to address these limitations by demonstrating the general utility of a hybrid approach that incorporates human coders to take advantage of the efficiencies machine learning techniques provide researchers when working with big datasets while still accounting for contextual nuance from using qualitative approaches. Even though the definition of misinformation may vary in other discussions (e.g., politics, climate change, other social issues), the general principles of the methodology described could be leveraged to provide more up-to-date and richer contextual insights into how these conspiracy-related discourses evolve over time p38, ¶2

5. Please elaborate on the relativistic approach: define it and justify its use in the current context.

Author Response: Thank you for your comment about expanding on the relativistic approach mentioned in our study. We have included additional language to the manuscript to further define this approach as well as including examples for how this approach can be adapted for other discourses.

Interpreting sentiment scores can be further complicated when accounting for the fact that the discussion topic can also influence the average emotional tone of a discourse. For example, discussions about a pandemic may have higher percentages of negative affect words (e.g., “death”, “tragedy”) compared to lighter conversation topics such as gardening. For this reason it is difficult to determine what is an appropriate threshold for meaningful sentiment scores across topics. While this can limit what researchers can infer from these analyses, this study addresses this limitation by using a relativistic interpretation of sentiment scores. More specifically, discourse will be marked as high in a sentiment category based on whether it is in the 90th percentile of scores within the corpus. Using the 90th percentile as a threshold marker allows us to account for the specific context of the 5G discourse when making judgements for determining what is a high level of sentiment. This approach is also adaptable across a wide array of topics since percentiles indicate which cases are high and low for a given metric based on the sample distribution. To further illustrate this point, a post containing 5% of death-related words may be in the 95th percentile for discourse about gardening, making it a “high” amount, but be within the 50th percentile for pandemic-related discourse, making it a typical percentage within the context of that corpus. Additionally, this study incorporates a qualitative coding approach to account for contextual information that is typically overlooked in sentiment analysis but can be recognized by a human coder. p8, ¶2

6. The first paragraph on page 9 appears disjointed as it initially discusses three modes of data interpretation, then transitions to discussing inductive and deductive analysis. It would be beneficial to establish a coherent connection between these two sets of approaches to ensure the flow of ideas and the logical progression of the argument.

Author Response: Thank you for your comment about the modes of data interpretation discussed in the manuscript. Our initial intent when discussing these modes were to further illustrate the advantages of human coders when interpreting data, however, we realize that this could be interpreted as introducing a formal coding approach for analysis. In order to make this clearer, we have edited the language in this paragraph to more explicitly emphasize the utility of human coders when identifying emerging narratives compared to machine learning approaches. 

1.3 Inductive and Deductive approaches for content coding

In recent years there have been calls for researchers to recognize the impact of contextualization when interpreting data, such as accounting for changes in semantics within a dataset over time (Poirier, 2021). When identifying emerging narratives within online discourse, where meanings of text vary due to novel co-occurrences of words, the boundaries between noise and relevant signal are constantly shifting. Compared to machine learning approaches, which requires an existing training dataset of annotated posts to be effective, human coders are more adaptable at updating their background knowledge of events, making them effective at recognizing text containing previously undocumented narratives. To account for contextualization factors, our study utilized both inductive and deductive analytic techniques to examine tweets related to conspiracy discourse that states a relationship between COVID-19 and 5G technology. p9, ¶1

7. I guess most of the audience of this paper should get some foundational knowledge, like what inductive and deductive coding are, as well as their pros and cons. So the authors could cut this part.

Author Response: Thank you for your comment about the inductive and deductive coding sections of our manuscript. We have edited the language in these sections to make these sections more concise while still providing a description of these approaches. 

Based in grounded theory (Glaser and Strauss, 1967), inductive coding is an iterative data analytic process centered on constant examination and comparison, which allows for theory development and explicit coding procedures (Corbin and Strauss, 1990; Boyatzis 1998). The primary benefits of an inductive approach is that it allows researchers to code texts using labels that are both aligned with the data and free from the influence of extant concepts, and detect tacit elements or connotations of the data that may not be apparent from a superficial reading of denotative content (Suddaby, 2006). For the current study, inductive coding was used to identify narratives and rumors prominent within the 5G discourse. 

Deductive coding, on the other hand, refers to a top-down coding process intended on testing whether data coincide with existing assumptions, theories, or hypotheses (Fereday & Muir-Cochrane, 2006). For the deductive coding scheme in the current study, we coded for whether posts contained misinformation or misinformation corrections based on whether it made statements claiming that 5G wireless technology causes COVID-19, or actively refutes the conspiracy. The criteria for classifying 5G-related misinformation and corrections was adapted from previous frameworks identifying COVID-related misinformation (Haupt et al., 2021a; Islam et al., 2020; Mackey et al., 2021). The researchers also coded for whether tweets expressed a positive, negative, or neutral stance towards 5G conspiracies. Coding for the user’s stance makes it possible to trace and compare general sentiment across topics without having to account for specific themes. p9, ¶2-3

8. Deductive coding is a coding process intended to test whether data coincides with existing assumptions, theories, or hypotheses. I agree with that. But I don’t see any assumptions or hypotheses that the authors sought to test.

Author Response: Thank you for your comment about clarifying the existing assumptions and theories associated with our deductive coding approach. We have edited this section to make our coding criteria clearer as well as made it more explicit that we adapted existing frameworks for defining misinformation from previous COVID-related studies.

Deductive coding, on the other hand, refers to a top-down coding process intended on testing whether data coincide with existing assumptions, theories, or hypotheses (Fereday & Muir-Cochrane, 2006). For the deductive coding scheme in the current study, we coded for whether posts contained misinformation or misinformation corrections based on whether it made statements claiming that 5G wireless technology causes COVID-19, or actively refutes the conspiracy. The criteria for classifying 5G-related misinformation and corrections was adapted from previous frameworks identifying COVID-related misinformation (Haupt et al., 2021a; Islam et al., 2020; Mackey et al., 2021). The researchers also coded for whether tweets expressed a positive, negative, or neutral stance towards 5G conspiracies. Coding for the user’s stance makes it possible to trace and compare general sentiment across topics without having to account for specific themes. p9, ¶3

9. My understanding is that this study summarized the topic and linguistic characteristics of 5G-related conspiracies on Twitter using topic modeling, LIWC, and manual content analysis. First, it is not about detection. The paper did not present any algorithms or computations that could be used to detect falsehoods in the future. Second, these methods are not quite new (see below)

Rashkin, H., Choi, E., Jang, J. Y., Volkova, S., & Choi, Y. (2017, September). Truth of varying shades: Analyzing language in fake news and political fact-checking. In Proceedings of the 2017 conference on empirical methods in natural language processing (pp. 2931-2937).

Zhou, C., Li, K., & Lu, Y. (2021). Linguistic characteristics and the dissemination of misinformation in social media: The moderating effect of information richness. Information Processing & Management, 58(6), 102679.

Massey, P. M., Kearney, M. D., Hauer, M. K., Selvan, P., Koku, E., & Leader, A. E. (2020). Dimensions of misinformation about the HPV vaccine on Instagram: Content and network analysis of social media characteristics. Journal of medical Internet research, 22(12), e21451.

Author Response: Thank you for your comment about clarifying the impact and potential novelty of our study. In response we have added additional language to our introduction to clarify that we are incorporating existing methodologies into a streamlined approach in order to conduct more in-depth characterization (versus “identification”) of emerging narratives in online discourse and have made this distinction consistent throughout the manuscript. We have also included the above mentioned references as examples of work that use these methods. We also clarify that by “detection”, this word can refer to a recognition process that is accessible to both humans and machine learning classifier models. This is further illustrated by the fact that the word “detect” has existed long before machine learning techniques were prevalent. Further, we emphasize that the approach described in this study can be used to streamline annotation of posts to develop relevant training datasets, which is a crucial step that needs to be taken before it’s possible to develop effective classifier models for future conspiracy discourse. Changes are numerous, please find some of the main changes addressing this comment below:

To gain a more thorough understanding of the nuanced patterns and dynamics underlying the spread of misinformation, our study introduces a hybrid analytic approach using metadata from 5G-COVID conspiracy discourse on Twitter (currently known as ‘X’ but referred to as Twitter for this paper) aimed at leveraging both the efficiency of NLP techniques and the qualitative schema afforded by human coders. More specifically, this study used topic modeling and sentiment analysis to identify influential posts and characterize the discourse, and then used both inductive and deductive coding to detect context-specific narratives that would not be measured by standard sentiment dictionaries. While NLP and qualitative coding methods have been widely used throughout the literature for identifying and characterizing misinformation (see the following for examples: Haupt et al., 2021a; Rashkin et al., 2017; Massey et al., 2020; Zhou et al., 2021), this paper combines these methods into a streamlined approach that can be utilized for rapidly characterizing nuanced themes and emerging narratives within large scale online discourses that requires significantly less burden on human annotation, as illustrated in the flow diagram in S1 Appendix. The hybrid method utilized in this study also has the potential to generate nuanced training data for machine learning classifier models without the need for annotating thousands of posts.

This study aims to further bridge the gap between computational and qualitative methodologies by demonstrating how NLP and manual annotation can be used synergistically to emphasize the positive aspects of each approach while minimizing their respective drawbacks. In order to assess the utility of this methodology, themes generated by this approach will be compared for consistency with 5G-COVID conspiracy themes previously identified on Twitter (Ahmed et al., 2020; Flaherty et al., 2022; Langguth et al., 2022), Facebook (Bruns et al., 2020), and Instagram (Quinn et al., 2021). Further, this study will extend the current literature by providing in-depth characterization of discourse focusing on correcting false information, which can be used to inform counter strategies for online misinformation propagation p4-5

---

## [Decision Letter · Decision Letter 1]

22 Nov 2023

Detecting Nuance in Conspiracy Discourse: Advancing Methods in Infodemiology and Communication Science with Machine Learning and Qualitative Content Coding

PONE-D-23-14668R1

Dear Dr. Haupt,

We’re pleased to inform you that your manuscript has been judged scientifically suitable for publication and will be formally accepted for publication once it meets all outstanding technical requirements.

Kind regards,

Stefano Cresci

Academic Editor

PLOS ONE

Additional Editor Comments (optional):

Thank you for the thorough work carried out as part of this revision. The reviewers are now fully satisfied with your work and the state of the manuscript. All concerns appear to be adequately addressed, including those from Reviewer #4 who was unavailable for this second round of review. In light of this revision, I can recommend publication.

Reviewers' comments:

Reviewer's Responses to Questions

**Comments to the Author**

1. If the authors have adequately addressed your comments raised in a previous round of review and you feel that this manuscript is now acceptable for publication, you may indicate that here to bypass the “Comments to the Author” section, enter your conflict of interest statement in the “Confidential to Editor” section, and submit your "Accept" recommendation.

Reviewer #1: All comments have been addressed

Reviewer #2: All comments have been addressed

Reviewer #3: All comments have been addressed

2. Is the manuscript technically sound, and do the data support the conclusions?

Reviewer #1: Yes

Reviewer #2: (No Response)

Reviewer #3: Yes

3. Has the statistical analysis been performed appropriately and rigorously? 

Reviewer #1: Yes

Reviewer #2: (No Response)

Reviewer #3: I Don't Know

4. Have the authors made all data underlying the findings in their manuscript fully available?

Reviewer #1: Yes

Reviewer #2: (No Response)

Reviewer #3: Yes

5. Is the manuscript presented in an intelligible fashion and written in standard English?

Reviewer #1: Yes

Reviewer #2: (No Response)

Reviewer #3: Yes

6. Review Comments to the Author

Reviewer #1: The manuscript presented in an intelligible fashion and written in standard English is a testament to the author's meticulousness and commitment to professionalism. From the very first line, clarity and coherence reign supreme, allowing readers to effortlessly follow the author's train of thought. The use of standard English further enhances the manuscript's accessibility, ensuring that it is easily readable to a wide range of audiences. The choice of appropriate language, precise grammar, and coherent structure all contribute to this professional tone. This manuscript showcases the author's dedication to delivering a high-quality piece of work, reflecting their expertise and attention to detail. It is a shining example of how effective communication and adherence to language standards can elevate the impact and credibility of any written document.

Reviewer #2: (No Response)

Reviewer #3: (No Response)

7. PLOS authors have the option to publish the peer review history of their article (what does this mean?). If published, this will include your full peer review and any attached files.

Reviewer #1: **Yes: **Abdulhamid Abuaniza

Reviewer #2: No

Reviewer #3: No

---

## [Editor Report · Acceptance letter]

11 Dec 2023

PONE-D-23-14668R1 

Detecting Nuance in Conspiracy Discourse: Advancing Methods in Infodemiology and Communication Science with Machine Learning and Qualitative Content Coding 

Dear Dr. Haupt:

I'm pleased to inform you that your manuscript has been deemed suitable for publication in PLOS ONE. Congratulations! Your manuscript is now with our production department. 

Kind regards, 

on behalf of

Dr. Stefano Cresci 

Academic Editor

PLOS ONE